# Assisted Learning for Organizations with Limited Imbalanced Data

**Cheng Chen**[*]                                                              *u0952128@utah.edu*
*Department of Electrical and Computer Engineering*
*University of Utah*
*Salt Lake City, UT 84112*

**Jiaying Zhou**[*]                                                            *zhou1054@umn.edu*
*School of Statistics*
*University of Minnesota*
*Minneapolis, MN 55455*

**Jie Ding**                                                                   *dingj@umn.edu*
*School of Statistics*
*University of Minnesota*
*Minneapolis, MN 55455*

**Yi Zhou**                                                                    *yi.zhou@utah.edu*
*Department of Electrical and Computer Engineering*
*University of Utah*
*Salt Lake City, UT 84112*

**Reviewed on OpenReview:** *https://openreview.net/forum?id=SEDWlhcFWA*

## Abstract

In the era of big data, many big organizations are integrating machine learning into their work pipelines to facilitate data analysis. However, the performance of their trained models is often restricted by limited and imbalanced data available to them. In this work, we develop an assisted learning framework for assisting organizations to improve their learning performance. The organizations have sufficient computation resources but are subject to stringent data-sharing and collaboration policies. Their limited imbalanced data often cause biased inference and sub-optimal decision-making. In assisted learning, an organizational learner purchases assistance service from an external service provider and aims to enhance its model performance within only a few assistance rounds. We develop effective stochastic training algorithms for both assisted deep learning and assisted reinforcement learning. Different from existing distributed algorithms that need to transmit gradients or models frequently, our framework allows the learner to only occasionally share information with the service provider, but still, obtain a model that achieves near-oracle performance as if all the data were centralized.

## 1 Introduction

Over the past decade, machine learning has demonstrated its great success in various engineering and science domains, e.g., robotic control (Kober & Peters, 2014; Deisenroth et al., 2013), natural language processing (Li et al., 2016; Bahdanau et al., 2017), computer vision (Liu et al., 2017; Brunner et al., 2017), finance (Lee et al., 2020; Lussange et al., 2021; Koratamaddi et al., 2021). Following this trend, many organizations, e.g., governments, hospitals, schools, and companies, are integrating machine learning models

---

[*]Equal contribution

into their work pipelines to facilitate data analysis and improve decision-making. For example, according to a recent survey (Financesonline, 2021), about 49% of companies worldwide are considering to use machine learning, 51% of organizations claim to be early adopters of machine learning, and the estimated productivity improvement obtained by learning models can be as high as 40%.

Although machine learning techniques have been standardized and are relatively easy for organizations to implement with real-world data, the model performance critically depends on the quality of the training data they hold (Goodfellow et al., 2016). However, these training data may be of limited size or biased toward certain distributions. For example, for local banks collecting financial data from economic activities, the sample size is often limited by the size of the local population, and the data distribution is affected by the local economy. Consequently, machine learning models trained on such limited and imbalanced data may generalize poorly on test data. Therefore, there is a need to develop a modern machine learning framework that can assist organizations in improving their model performance.

A natural solution is to connect the organization to an external consulting agent, who has machine learning expertise and holds a large amount of data that is balanced or complementary to the organization's imbalanced data. For example, for a local small hospital, the consulting agent may be a large medical center at the state or national level. In particular, the organization can purchase some generic assistance service (e.g., information exchange) from the consulting agent to help improve its model performance. However, organizations' interactions with external consulting agents are usually subject to stringent regulations and policies. Two common restrictions are: (i) neither the organization nor the consulting agent wants to share their raw data with the other side; and (ii) the organization often has a limited budget and can purchase very limited services from the agent. Considering these challenges for organizational machine learning users, it is desired to develop a machine learning framework for organizations to significantly improve their model performance by purchasing limited assistance services from external agents without data sharing. This constitutes the overarching goal of this work.

In this work, we develop an assisted learning framework in the "horizontal-splitting" setting, where the organization and the consulting agent possess different sets of data samples that are utilized for training a common model. In our context, the organization's data are assumed to be limited and imbalanced, while the consulting agent's data are large or complement the organization's data. Our assisted learning framework is inspired by organizations' common characteristics: they may have a limited budget for the rounds of purchasing external assistance, yet they can exchange a large amount of side information per assistance round to maximize the performance gain. We summarize our contributions as follows.

## 1.1 Our Contributions

We formally define an assisted learning framework with 'horizontal-splitting' data. In this framework, an organization (referred to as the learner) is connected to an external service provider (referred to as the provider) for assistance, and their interaction is subject to the following rules: (i) Neither the learner nor the provider can share data with each other; (ii) The learner can interact with the provider for only a few assistance rounds due to limited budget, and it aims to maximize the performance gain; and (iii) In each assistance round, both the learner and the provider have access to sufficient computation and communication resources.

We then develop a fully-decentralized assisted learning algorithm named AssistDeep for *deep learning* tasks. This algorithm enables the learner to effectively interact with the provider without sharing data. Specifically, every assistance round of AssistDeep consists of two phases. In the first phase, the learner performs local training for multiple iterations and sends the generated trajectory of models together with their corresponding local loss values to the provider. In the second phase, the provider utilizes the learner's information to evaluate the global loss of the received models and picks the model that achieves the minimum global loss as the initialization. Then, the provider performs local training for multiple iterations and sends the generated trajectory of models together with their corresponding local loss values to the learner. Finally, the learner utilizes the provider's information to evaluate the global loss of the received models and outputs the best model that achieves the minimum global loss. Moreover, for *reinforcement learning* tasks, we further propose

a fully-decentralized assisted learning algorithm named AssistPG, which is based on the policy gradient algorithm and has the same training logic as that of AssistDeep.

Through extensive experiments on deep learning and reinforcement learning tasks, we demonstrate that the learner can achieve a near-oracle performance with AssistDeep and AssistPG as if the model was trained on centralized data (i.e., the learner has full access to the provider's raw data). In particular, as the learner data's level of imbalance increases, AssistDeep can help the learner achieve a higher performance gain. Moreover, data are never exchanged in the assisted learning process for both participants.

## 1.2 Related Work

**Assisted learning**. The concept of assisted learning was originally proposed by the earlier work (Xian et al., 2020). However, they consider a very different assisted learning setting, i.e., the 'vertical-splitting' setting in which the organization and the consulting agent collect different features from the same cohort. This is in contrast with our setting where they hold the same features but different data samples. Moreover, their assisted learning algorithm was specially developed for regression-type tasks, whereas our algorithms apply to both classification and reinforcement learning tasks with general deep models. Consequently, our algorithm designs and application scenarios are substantially different from the prior work.

**Distributed learning**. Distributed learning frameworks such as federated learning (Shokri & Shmatikov, 2015; Konecny et al., 2016; McMahan et al., 2017) aim to improve the learning performance for a large number of learners that have limited data and computation/communication resources. These learning frameworks are well suited for cloud systems and IoT systems (Ray, 2016; Gomathi et al., 2018) that manage numerous smart devices through wireless communication. In conventional distributed optimization, the data is evenly distributed among workers, which collaboratively solves a large-scale problem by frequently exchanging local information (i.e., gradients or models) via either decentralized networks (Xie et al., 2016; Lian et al., 2017; 2018) or centralized networks (Ho et al., 2013; Li et al., 2014; Richtarik & Takavc, 2016; Zhou et al., 2016; 2018). As a comparison, our assist learning framework requires only a few interaction rounds between the learner and provider. This is particularly appealing for organizational learners, who can employ a sophisticated optimization process locally while restricting the rounds of assistance.

# 2 Assisted Deep Learning

In this section, we introduce the assisted learning framework for deep learning tasks. Throughout the paper, L denotes a learner who seeks assistance, and P denotes a service provider who provides assistance to L.

## 2.1 Problem Formulation

We consider the case where the learner L aims to train a machine learning model $\theta$ that performs well on its own dataset $\mathcal{D}^{(\mathrm{L})}$ and generalizes well to unseen data. In general, L can train a machine learning model by solving the empirical risk minimization problem $\min_{\theta \in \Theta} f(\theta; \mathcal{D}^{(\mathrm{L})})$, where $f(\cdot; \mathcal{D}^{(\mathrm{L})})$ is the loss on $\mathcal{D}^{(\mathrm{L})}$ and $\Theta$ is the parameter space. Standard statistical learning theories show that the obtained model can generalize well to intact test samples under suitable constraints of model parsimoniousness (Ding et al., 2018). However, when the learner's data $\mathcal{D}^{(\mathrm{L})}$ contains a **limited** number of samples that are highly **imbalanced**, the learned model will suffer from overfitting or deteriorated generalization capability to the unseen test data.

To overcome data deficiency, the learner L intends to connect with an external service provider P (e.g., a commercialized data company), who possesses data $\mathcal{D}^{(\mathrm{P})}$ that are sufficient or complementary to the learner's data $\mathcal{D}^{(\mathrm{L})}$. Ideally, the learner L would improve the model by solving the following data-augmented problem, where $\mathcal{D}^{(\mathrm{L,P})} := \mathcal{D}^{(\mathrm{L})} \cup \mathcal{D}^{(\mathrm{P})}$ denotes the centralized data.

$$\theta^{(\mathrm{L,P})} = \arg\min_{\theta \in \Theta} f(\theta; \mathcal{D}^{(\mathrm{L,P})}). \tag{1}$$

We note that $f(\theta; \mathcal{D}^{(\mathrm{L,P})}) = f(\theta, \mathcal{D}^{(\mathrm{L})}) + f(\theta, \mathcal{D}^{(\mathrm{P})})$. If $\mathcal{D}^{(\mathrm{P})}$ is generated from a distribution that is close to the underlying data distribution, it is expected that $\theta^{(\mathrm{L,P})}$ will achieve significantly improved performance on

---

**Algorithm 1** AssistDeep

---

**Input:** Initialization model $\theta^0$, learning rate $\eta$, assistance rounds $R$, number of local iterations $T$ (for learner) and $T'$ (for provider).

**for** assistance rounds $r = 1, \ldots, R$ **do**

    **Learner L :**
        ▶ Initialize $\theta_0^{(\mathrm{L})} = \theta^{r-1}$
        ▶ Local training to generate $\{\theta_t^{(\mathrm{L})}\}_{t=0}^{T-1}$
        ▶ Send $\{\theta_t^{(\mathrm{L})}, f(\theta_t^{(\mathrm{L})}; \mathcal{D}^{(\mathrm{L})})\}_{t \in \mathcal{T}}$ to provider P

    **Provider P :**
        ▶ Initialize $\theta_0^{(\mathrm{P})} = \arg\min_{\theta \in \{\theta_t^{(\mathrm{L})}\}_{t \in \mathcal{T}}} f(\theta; \mathcal{D}^{(\mathrm{L,P})})$
        ▶ Local training to generate $\{\theta_t^{(\mathrm{P})}\}_{t=0}^{T'-1}$
        ▶ Send $\{\theta_t^{(\mathrm{P})}, f(\theta_t^{(\mathrm{P})}; \mathcal{D}^{(\mathrm{P})})\}_{t \in \mathcal{T}'}$ to learner L

    **Learner L :**
        ▶ Output $\theta^r = \arg\min_{\theta \in \{\theta_t^{(\mathrm{P})}\}_{t \in \mathcal{T}'}} f(\theta; \mathcal{D}^{(\mathrm{L,P})})$

**end**

**Output:** The best model in $\{\theta^r\}_{r=1}^{R}$

---

unseen data. However, it is unrealistic to centralize the data since the interactions between the learner L and the provider P are often restricted by stringent regulations. More specifically, in the assisted learning framework, we consider the following protocols listed below.

---

### Assisted Learning Protocols

1. *No data sharing:* Neither the learner L nor the provider P will share data with each other.

2. *Limited assistance:* The learner L has a limited budget for purchasing assistance services. The learner desires to maximize the performance gain within only a few assistance rounds.

3. *Unlimited computation and communication:* In each assistance round, both the learner and the provider can perform unlimited computation and exchange unlimited information.

---

To elaborate on the above protocols, first note that data sharing is often considered sensitive and prohibited in modern distributed learning. Also, assistance service between organizations is usually costly and time-consuming in reality, and organizational learners usually have a limited budget to purchase and manage such service. Moreover, we generally assume that both the organizational learner and the service provider have sufficient computation resources, and they can exchange unlimited information in each interaction round, e.g., the learner (resp. provider) can send an employee (resp. technician) to deliver a large-capacity hard drive to the other side.

Conventional distributed learning algorithms cannot be applied to enable assisted learning, as they are developed for distributed systems involving a large number of agents having access to very limited computation and communication resources, and they often require frequent information exchange among the agents. Hence, we need to develop a training algorithm specifically for assisted learning that can substantially improve the learner's model performance via limited interactions with the service provider. Next, we present a generic assisted learning algorithm for solving deep learning tasks.

### 2.2 AssistDeep for Assisted Deep Learning

We propose *AssistDeep* in Algorithm 1 for assisted deep learning. The learning process consists of $R$ rounds, each consisting of the following interactions between the learner L and the provider P.

(1) First, the learner L initiates a local learning process. It initializes a model $\theta_0^{(\mathrm{L})}$ and applies any standard deep learning optimizer (e.g., SGD, Adam, etc.) with learning rate $\eta$ to update it for $T$ iterations using the local dataset $\mathcal{D}^{(\mathrm{L})}$. Then, the learner evaluates the local loss $f(\cdot; \mathcal{D}^{(\mathrm{L})})$ in a subset $\mathcal{T}$ of the iterations $t = 0, 1, \ldots T - 1$. Lastly, the learner sends this subset of models and their corresponding local loss to the provider P .

(2) Upon receiving the information from the learner L, the provider P first evaluates the global loss $f(\cdot; \mathcal{D}^{(\mathrm{L,P})})$ of the received set of models $\{\theta_t^{(\mathrm{L})}, t \in \mathcal{T}\}$ and picks the one that achieves the minimum global loss as the initialization model $\theta_0^{(\mathrm{P})}$. Note that the global loss can be evaluated because the local loss $\{f(\theta_t^{(\mathrm{L})}; \mathcal{D}^{(\mathrm{L})}), t \in \mathcal{T}\}$ are provided by the learner L , and the provider P just needs to evaluate the local loss $\{f(\theta_t^{(\mathrm{L})}; \mathcal{D}^{(\mathrm{P})}), t \in \mathcal{T}\}$ on its local data. After that, the provider applies any standard optimizer with learning rate $\eta$ to update the model for $T'$ iterations on the local dataset $\mathcal{D}^{(\mathrm{P})}$. Then, the provider evaluates the local loss $f(\cdot; \mathcal{D}^{(\mathrm{P})})$ in a subset $\mathcal{T}'$ of the iterations $t = 0, 1, \ldots T' - 1$, and sends the subset of models and their corresponding local loss to the learner L.

(3) Once the learner L receives the information sent by the provider P, it evaluates the global loss $f(\cdot; \mathcal{D}^{(\mathrm{L,P})})$ of the received set of models $\{\theta_t^{(\mathrm{P})}, t \in \mathcal{T}'\}$ and picks the one that achieves the minimum global loss as the output model of this assistance round.

**Discussions**. The above assisted learning algorithm works for general deep learning tasks. It does not require data sharing between the learner and the provider. In particular, the interaction between the learner and the provider has the following two prominent features.

- Both the learner and the provider need to store a number of models sampled from the trajectory of models generated in their local training process and evaluate the corresponding local loss value of these sampled models. Then, these sampled models and their local loss values are sent to the other side. Normally, this information exchange may require a large amount of communication. But as we show later in the experimental studies, it suffices to sample the training trajectory at a low frequency.

- After receiving the models and loss values sent by the other side, both the learner and the provider will evaluate the global loss of these models on the centralized data, and then will pick the one that achieves the minimum global loss as the initialization/output model. Here, the global loss of these models can be evaluated on one side, as the loss values of the models on the other side's local data are evaluated and sent by the other side.

### 2.3 Convergence Analysis of Assisted Learning

In this subsection, we show that assisted learning provably converges to a stationary point in smooth nonconvex optimization. For simplicity, we consider the full gradient setting, where the local training of AssistDeep uses full gradient updates. We also make the following standard assumptions.

**Assumption 1.** We assume that the assisted learning problem in the goal (1) satisfies the following conditions.

1. The global loss $f(\theta; \mathcal{D}^{(\mathrm{L,P})})$ has $L$-Lipschitz gradients. Moreover, $\inf_\theta f(\theta; \mathcal{D}^{(\mathrm{L,P})}) > -\infty$;
2. There exists a constant $G > 0$ such that $\max\{\|\nabla f(\theta; \mathcal{D}^{(\mathrm{L})})\|, \|\nabla f(\theta; \mathcal{D}^{(\mathrm{P})})\|, \|\nabla f(\theta; \mathcal{D}^{(\mathrm{L,P})})\|\} \leq G$ for all $\theta$ generated by AssistDeep.

Note that in each assistance round $r$, both the provider and the learner pick the best model via the $\arg\min$ operation to initialize and finalize their local training. This guarantees that AssistDeep continuously makes optimization progress. Specifically, we let $\theta_0^{(\mathrm{L}),r}, \theta_0^{(\mathrm{P}),r}$ denote learner L's and provider P's initialization models, respectively, in the round $r$. Recall that $\theta^r$ is the output model of learner L (see Algorithm 1). Then, the two $\arg\min$ operations guarantee that the following proposition holds.

**Proposition 1.** The sequence of global loss $\{f(\theta^r; \mathcal{D}^{(\mathrm{L,P})})\}_r$ achieved by AssistDeep monotonically decreases, i.e.,

$$f(\theta^r, \mathcal{D}^{(\mathrm{L,P})}) \leq f(\theta_0^{(\mathrm{P}),r}, \mathcal{D}^{(\mathrm{L,P})}) \leq f(\theta_0^{(\mathrm{L}),r}, \mathcal{D}^{(\mathrm{L,P})}) = f(\theta^{r-1}, \mathcal{D}^{(\mathrm{L,P})}).$$

Next, we prove that the output model $\theta^r$ asymptotically converges to a stationary point. The full proof is presented in the appendix of this paper.

**Theorem 1.** Let Assumption 1 hold and run AssistDeep with full gradient updates for $R$ rounds. Choose learning rate $\eta = \mathcal{O}((RLTG^2)^{-0.5})$. Then, $\min_{0 \leq r \leq R-1} \|\nabla f_{\mathrm{L,P}}(\theta^r)\| \overset{R}{\to} 0$.

Therefore, with a proper choice of the learning rate $\eta$, assisted learning is guaranteed to find a stationary point in general nonconvex optimization. We note that it is challenging to establish such a convergence result for AssistDeep, as the model chosen by the arg min operations can be any model in the entire model trajectories. As we show in the experiments later, our AssistDeep can often achieve a similar performance to that of SGD with centralized data.

### 2.4 Comparison of Federated Learning and the Proposed Assisted Learning

Federated learning is a widely-adopted distributed learning framework (Shokri & Shmatikov, 2015; Konecny et al., 2016; McMahan et al., 2017; Zhao et al., 2018; Li et al., 2020; Diao et al., 2021; 2022). It focuses on learning a global model by averaging local models trained on numerous smart devices, showcasing substantial differences from assisted learning in terms of application scenarios and algorithms. Firstly, federated learning seeks to enhance learning performance for a multitude of learners with limited data and computation/communication resources, making it well-suited for cloud and IoT systems. Conversely, assisted learning strives to support a single organization-level learner (e.g., small labs or local clinics) in consistently and substantially improving learning performance through a limited number of interactions with an external organization-level service provider (e.g., research institutes or hospitals). Both learner and provider possess ample resources but are constrained by strict data-sharing policies. Secondly, existing federated learning algorithms necessitate frequent transmission and exchange of local information (i.e., gradients or model parameters). In comparison, assisted learning algorithms mandate only a few interaction rounds between the learner and provider. This approach is particularly attractive for organizational learners, as it allows them to utilize a refined optimization process locally while limiting assistance rounds to reduce the cost of acquiring services from external providers.

Moreover, in contrast to assisted learning (specifically, the AssistDeep algorithm), it is worth noting that the conventional federated learning framework, such as the FedAvg algorithm (McMahan et al., 2017), exhibits sensitivity to data heterogeneity and imbalance, even though FedAvg can address the same problem (involving two clients) as AssistDeep. To elaborate, during each round of FedAvg, the cloud server compiles all the latest local models generated by the clients to derive a global model. This approach is susceptible to both local data heterogeneity and over-fitting during local training. In comparison, in AssistDeep, both the learner and provider transmit their local models and corresponding loss values to the other party for evaluation, and the optimal local model that achieves the lowest global loss is selected as the initialization for the subsequent round. This filtering process is highly nonlinear, as opposed to the linear weighted combination utilized in FedAvg. To validate this, we examine the performance of FedAvg alongside that of our AssistDeep in experiments involving highly imbalanced data (see Appendix A.1). The results indicate that FedAvg converges slowly due to the significant data imbalance, while our AssistDeep demonstrates superior and more robust performance. This can be attributed to FedAvg only linearly aggregating models generated during the final iterations of local training, whereas AssistDeep conducts screening across all models produced during local training and selects the best one.

## 3 Assisted Reinforcement Learning

We further extend our assisted learning framework to Reinforcement Learning (RL) scenarios to enhance the model's generalizability. We first introduce some basic setups of RL.

**Markov Decision Process (MDP).** We consider a standard finite-horizon MDP that is denoted by a tuple $M = (\mathcal{S}, \mathcal{A}, \mathbf{P}, r, \pi, \rho_0, T)$, where $\mathcal{S}$ is the state space, $\mathcal{A}$ corresponds to the action space, $\mathbf{P} : \mathcal{S} \times \mathcal{A} \times \mathcal{S} \to [0, 1]$ denotes the underlying state transition kernel that drives the new state given the previous state and action, $r : \mathcal{S} \times \mathcal{A} \mapsto \mathbb{R}$ is the reward function, $\pi : \mathcal{S} \to \mathcal{A}$ is the policy, $\rho_0$ denotes the initial state distribution, and $T$ is the episode length. Given a policy $\pi_\theta$ parameterized with $\theta$, the goal of RL, also known as on-policy

learning, is to learn an optimal policy parameter $\theta^*$ that maximizes the expected accumulated reward, namely $\theta^* = \arg\max_\theta J(\theta) := \mathbb{E}[\sum_{t=1}^{T} \gamma^{t-1} r_t]$.

### 3.1 Problem Formulation

We assume that an RL learner L has collected a small amount of Markovian data $\mathcal{D}^{(L)}$ by interacting with a certain environment. It wants to train a policy that generalizes well to other similar environments. However, the data and environment that the learner L can access are limited. In assisted reinforcement learning, the learner L aims to enhance its policy's generalizability to unseen environments by querying assistance from a service provider P. For example, autonomous-driving startup companies typically own limited data that are insufficient for training good autonomous driving models that perform well in heterogeneous environments, and they can purchase assistance services from big companies (who own massive data) to improve the model performance and generalizability.

Formally, we assume that there is an underlying distribution of transition kernel that models the variability of the environment. Specifically, denote $E_\beta$ as an environment with the transition kernel $\mathbf{P}_\beta$ parameterized by $\beta$, which follows an underlying distribution $q$. Let $J_\beta(\theta)$ denote the expected accumulated reward collected from the environment $E_\beta$ following the policy $\pi_\theta$. The learner L's ultimate goal is to learn a good policy that applies to the underlying distribution of environment, namely, $\max_\theta \mathbb{E}_{\beta \sim q}[J_\beta(\theta)]$. In practice, the learner L only has training data collected from a limited number of environment instances, say $\beta^{(L)} = \{\beta_1^{(L)}, \ldots, \beta_{n_L}^{(L)}\}$. On the other hand, the service provider may have rich experience interacting with a more diverse set of environments, say $\beta^{(P)} = \{\beta_1^{(P)}, \ldots, \beta_{n_P}^{(P)}\}$. Consequently, the learner aims to solve the following RL problem with centralized data.

$$\max_\theta J_{\beta^{(L,P)}}(\theta) := \sum_{\beta \in \beta^{(L)}} J_\beta(\theta) + \sum_{\beta' \in \beta^{(P)}} J_{\beta'}(\theta). \tag{2}$$

### 3.2 AssistPG for Assisted Reinforcement Learning

Policy gradient (PG) is a classic RL algorithm for policy optimization. The PG algorithm estimates the policy gradient $\nabla J(\theta)$ via the policy gradient theorem, and applies it to update the policy. Specifically, given one episode $\tau$ with length $T$ that is collected under the current policy $\pi_\theta$, the corresponding policy gradient takes the following form, where $R(\tau) = \sum_{t=1}^{T} \gamma^{t-1} r_t$ is the discounted accumulated reward over this episode. In practice, a mini-batch of episodes is used to estimate the policy gradient (Sutton & Barto, 2018), which is approximated as follows.

$$\nabla J(\theta) \approx R(\tau) \sum_{t=1}^{T} \nabla \log \pi_\theta(a_t^{(i)} | s_t^{(i)}).$$

In Algorithm 2 below, we present **Assisted Policy Gradient (AssistPG)**–a policy gradient-type algorithm for solving the assisted RL problem in Equation (2). The main logic of the AssistPG algorithm is the same as that of the AssistDeep for assisted deep learning.

## 4 Experiments

In this section, we first visualize AssistDeep training to help understand the mechanism of assisted learning. Then, we provide extensive experiments of deep learning and reinforcement learning to demonstrate the effectiveness of the proposed assisted learning algorithms.

### 4.1 Visualization of AssistDeep Training

**Regression Example.** We apply AssistDeep to solve a regression problem with simulated data $a = [-1, -1]$, $b = [1, -1.25]$ and hyperparameters $T = 10$, $\eta = 0.9^r$ for both the learner and the provider. We run the algorithm for $R = 10$ assistance rounds. Fig. 1 shows the learning trajectory of $\theta^r$ for $r = 0, 1, \ldots, 9$. It can

---

**Algorithm 2** AssistPG

---

**Input:** Initialization model $\theta^0$, learning rate $\eta$, assistance rounds $R$, number of local iterations $T$ (for learner) and $T'$ (for provider).

**for** assistance rounds $r = 1, \ldots, R$ **do**

    **Learner L :**

        ▶ Initialize $\theta_0^{(\mathrm{L})} = \theta^{r-1}$

        ▶ Local PG training to generate $\{\theta_t^{(\mathrm{L})}\}_{t=0}^{T-1}$

        ▶ Send $\{\theta_t^{(\mathrm{L})}, \sum_{\beta \in \beta^{(\mathrm{L})}} J_\beta(\theta_t^{(\mathrm{L})})\}_{t \in \mathcal{T}}$ to provider P

---

    **Provider P :**

        ▶ Initialize $\theta_0^{(\mathrm{P})} = \arg\max_{\theta \in \{\theta_t^{(\mathrm{L})}\}_{t \in \mathcal{T}}} J_{\beta^{(\mathrm{L,P})}}(\theta)$

        ▶ Local PG training to generate $\{\theta_t^{(\mathrm{P})}\}_{t=0}^{T'-1}$

        ▶ Send $\{\theta_t^{(\mathrm{P})}, \sum_{\beta \in \beta^{(\mathrm{P})}} J_\beta(\theta_t^{(\mathrm{P})})\}_{t \in \mathcal{T}'}$ to learner L

---

    **Learner L :**

        ▶ Output $\theta^r = \arg\max_{\theta \in \{\theta_t^{(\mathrm{P})}\}_{t \in \mathcal{T}'}} J_{\beta^{(\mathrm{L,P})}}(\theta)$

**end**

**Output:** The best model in $\{\theta^r\}_{r=1}^R$

---

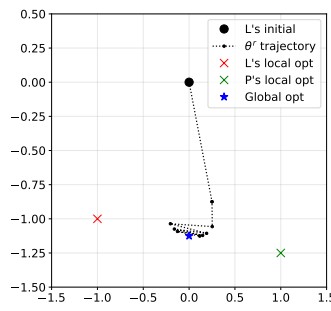

Figure 1: Learning trajectory of AssistDeep in a synthetic regression example.

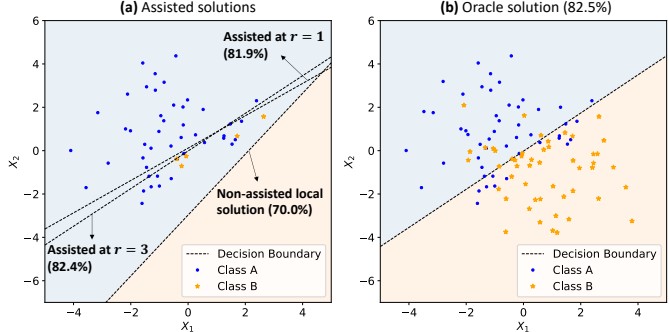

Figure 2: Visualization of AssistDeep in classification: (a) the learner's classifiers after being assisted by the provider at different rounds, and (b) oracle classifier obtained by using centralized data. The test accuracies are shown in the parentheses.

be seen that at the beginning, the learner L's learning trajectory moves toward the oracle solution since the directions of two local optima are roughly the same; then, it oscillates in between two opposite directions and converges to the oracle solution.

**Classification Example.** We apply AssistDeep to solve a simple logistic regression problem for binary classification. We generate two classes of data samples: Class A data contains 50 points drawn from $\mathcal{N}([-1, 1], 1.5^2 I_2)$, and Class B data contains 50 points drawn from $\mathcal{N}([1, -1], 1.5^2 I_2)$, where $\mathcal{N}$ and $I$ denote Gaussian distribution and identity matrix, respectively. Suppose that a learner L observes 90% class A samples and 10% class B samples. Another service provider P observes a similar number of data samples consisting of 10% class A samples and 90% class B samples. The learning process of AssistDeep is illustrated in Fig. 2 (Left), and we also show the oracle solution trained by SGD with centralized data in Fig. 2 (Right).

It can be seen that without any assistance, the learner can only achieve 70% accuracy and the corresponding classifier performs poorly on the samples in class B. In comparison, after a single round of assistance, the classification performance is significantly improved to 81.9% accuracy. After three rounds of assistance, the corresponding classifier is relatively close to the oracle classifier, which achieves an accuracy of 82.5%. Hence, it can be seen that AssistDeep has the potential to achieve a near-oracle performance.

## 4.2 Assisted Deep Learning Experiments

We test the performance of AssistDeep by comparing it with two baselines: SGD (using centralized data $\mathcal{D}^{(\mathrm{L,P})}$) and Learner-SGD (using only the learner's data $\mathcal{D}^{(\mathrm{L})}$). We consider two popular datasets CIFAR-10 (Krizhevsky, 2009) and SVHN. We implement all these algorithms using the Adam optimizer with learning rate 0.001 and batch size 256 to train AlexNet and ResNet-18 on CIFAR-10 and SVHN, respectively.

To test AssistDeep, we split the classification dataset into two parts and assign them to the learner and the provider, respectively. Specifically, the provider's data consists of half of the total samples of each classification class, and therefore is balanced and of a large size. On the other hand, the learner's data is sampled from the rest half of the data (excluding the provider's data) and is determined by the data imbalance level parameter $\gamma_L := |\mathcal{D}^{(\mathrm{LMINOR})}|/|\mathcal{D}^{(\mathrm{LMAJOR})}| \in [0,1]$, where $\mathcal{D}^{(\mathrm{LMAJOR})}$ and $\mathcal{D}^{(\mathrm{LMINOR})}$ denote the major class and minor class of the learner's local data, respectively, and we fix the major class to include the remaining data samples of the smallest class. Then, all the other classes are minor class of the learner's data and their sizes are given by $\gamma_L |\mathcal{D}^{(\mathrm{LMAJOR})}|$. Intuitively, $\gamma_L = 1$ means that learner's data is balanced and large, whereas $\gamma_L = 0$ means that learner's data is extremely imbalanced and small.

We fix the number of assistance rounds to be 10. The total number of local optimization iterations in each assistance round is fixed to be 2000. We assign these iteration budget to the learner and provider in proportion to their local sample sizes. Both the learner and provider record their local training models and local loss values for every $I = 50$ iteration, which is referred to as the sampling period. In addition, we conducted repeated experiments with different random seeds to estimate the standard deviation of the above results. We find that the training loss and the test accuracy of the output model are highly consistent across the repeated experiments. Specifically, the standard deviation of the training loss and test accuracy are 4e-3 and 3e-3, respectively. Therefore, in all the figures corresponding to deep learning experiments, we only plot the curves using the data obtained from one particular experiment, and the curves of all the other repeated experiments are nearly identical.

### 4.2.1 Effect of Sample Size and Data Imbalance Level

**CIFAR-10 Dataset.** We first compare these algorithms with balanced learner's data ($\gamma_L = 1$) and imbalanced learner's data ($\gamma_L = 0, 0.3, 0.7$) in training an AlexNet on the CIFAR-10 dataset. In Fig. 3, we plot the training loss (on centralized data $\mathcal{D}^{(\mathrm{L,P})}$) and the test accuracy (on the 10k test data) against the number of assistance rounds. Here, one assistance round on the x-axis is interpreted as 2k local iterations for SGD and Learner-SGD. The training loss of Learner-SGD is not reported as it is trained on $\mathcal{D}^{(\mathrm{L})}$ only. It can be seen that AssistDeep achieves a comparable performance to that of SGD with centralized data. In particular, when $\gamma_L = 0$, meaning that the learner has limited and extremely imbalanced data, the test performance of AssistDeep is significantly better than Learner-SGD, demonstrating the effectiveness of querying assistance from the service provider. When $\gamma_L = 0.3, 0.7, 1$ and the learner has more data, AssistDeep still achieves a near-oracle performance, and its test performance is better than Learner-SGD.

We further test and compare these algorithms with balanced learner's data $\gamma_L = 1$ and imbalanced learner's data $\gamma_L = 0, 0.3, 0.7$ in training a ResNet-18 on CIFAR-10. The results on ResNet-18 are shown in Fig. 4. It can be seen that in all the results, AssistDeep achieves a comparable test performance to that of SGD. Also, its test performance is better than Learner-SGD, especially when the learner has limited and imbalanced data ($\gamma_L = 0, 0.3, 0.7$). Combining these results with those in Fig. 3, we conclude that AssistDeep improves the test performance more (compare to Learner-SGD) when the learner's data is more imbalanced and limited.

**SVHN Dataset.** In Fig. 5 and 6, we repeat the above experiments with a different SVHN dataset using the same hyper-parameter settings. One can make the same observations from these results, which show that our proposed AssistDeep works well on diverse types of datasets.

### 4.2.2 Effect of Sampling Period

On CIFAR-10, we explore whether increasing the sampling frequency of the model and loss value can improve the performance of AssistDeep. We consider $\gamma_L = 0, 1$ and compare AssistDeep with centralized SGD under different sampling periods 20 and 50. The comparison results in training AlexNet and ResNet-18 are shown in

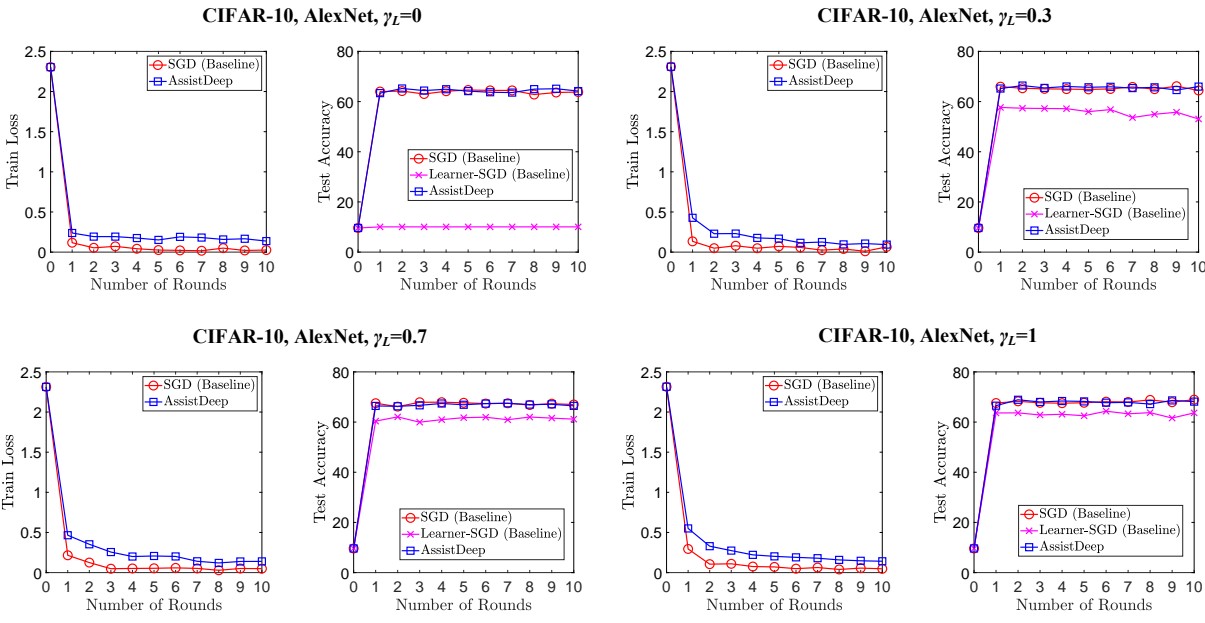

Figure 3: Comparison of AssistDeep, SGD, and Learner-SGD with $\gamma_L = 0, 0.3, 0.7, 1$ in training an AlexNet on CIFAR-10.

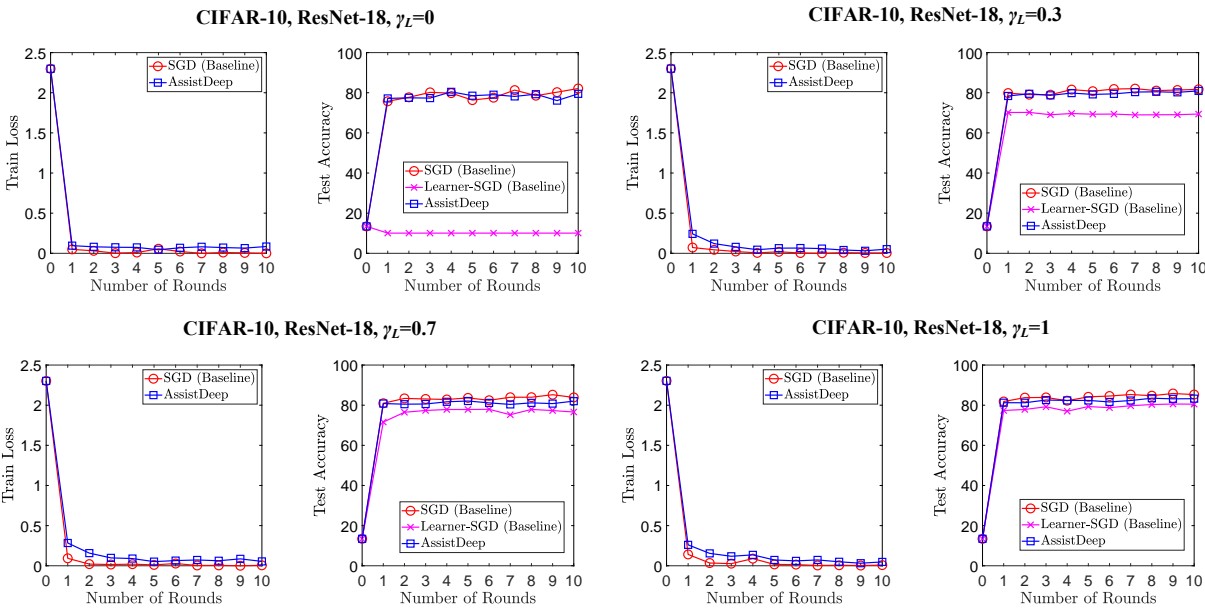

Figure 4: Comparison of AssistDeep, SGD, and Learner-SGD with $\gamma_L = 0, 0.3, 0.7, 1$ in training a ResNet-18 on CIFAR-10.

Fig. 7 and 8, respectively. It can be seen that using a low sampling frequency for AssistDeep already achieves the baseline performance of SGD. It implies that AssistDeep does not require much information exchange between the learner and provider. This helps save computation resources and reduce information leakage.

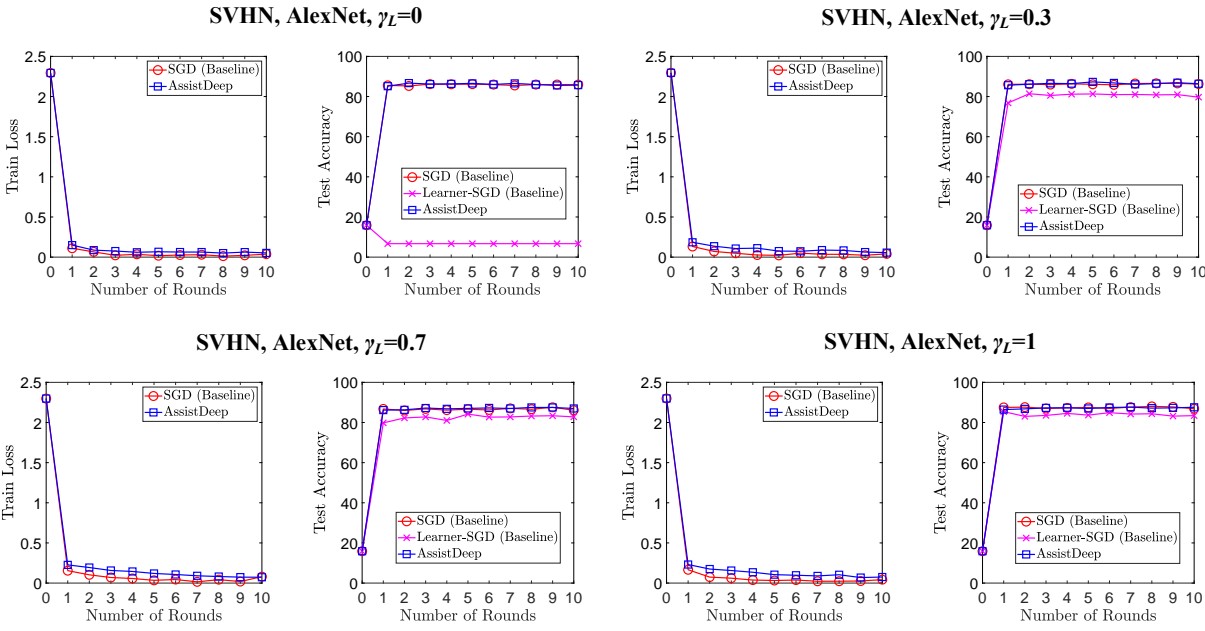

Figure 5: Comparison of AssistDeep, SGD, and Learner-SGD with $\gamma_L = 0, 0.3, 0.7, 1$ in training an AlexNet on SVHN.

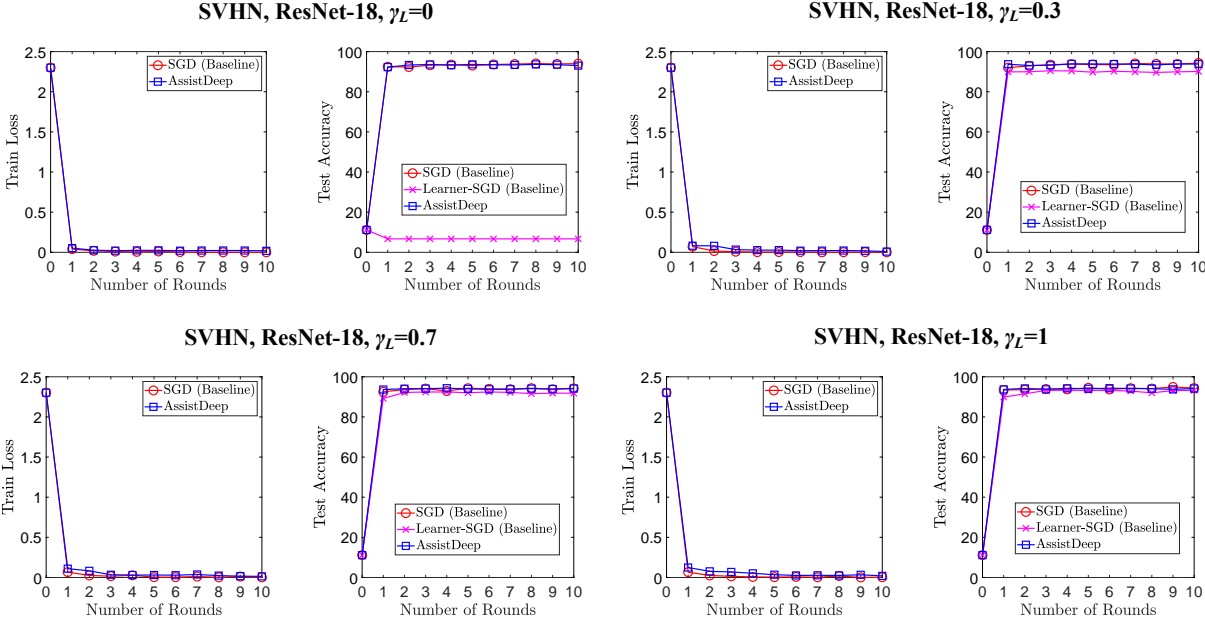

Figure 6: Comparison of AssistDeep, SGD, and Learner-SGD with $\gamma_L = 0, 0.3, 0.7, 1$ in training a ResNet-18 on SVHN.

### 4.3 Assisted Reinforcement Learning Experiments

We demonstrate the effectiveness of AssistPG via two RL applications: CartPole (Barto et al., 1983) and LunarLander (Brockman et al., 2016). In the CartPole problem, a controller aims to stabilize a pole attached to a cart by applying left or right force to the cart (see the first figure in Fig. 9), and we show that AssistPG can help the controller stabilize the pole with different pole lengths. For the LunarLander problem, a lander

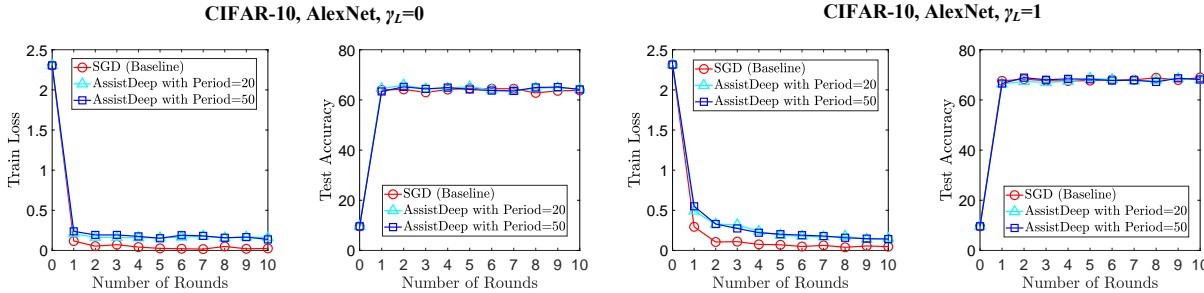

Figure 7: Comparison of AssistDeep and SGD with $\gamma_L = 0, 1$ under different sampling periods in training an AlexNet on CIFAR-10.

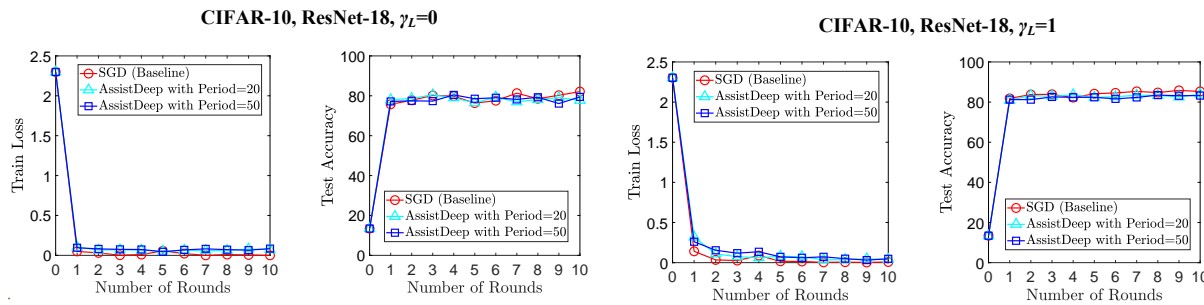

Figure 8: Comparison of AssistDeep and SGD with $\gamma_L = 0, 1$ under different sampling periods in training a ResNet-18 on CIFAR-10.

initializes its landing from the top left of the sky and aims to land on a landing pad by controlling its engine (see the first figure in Fig. 10). We show that AssistPG can help land the lander with different engine powers.

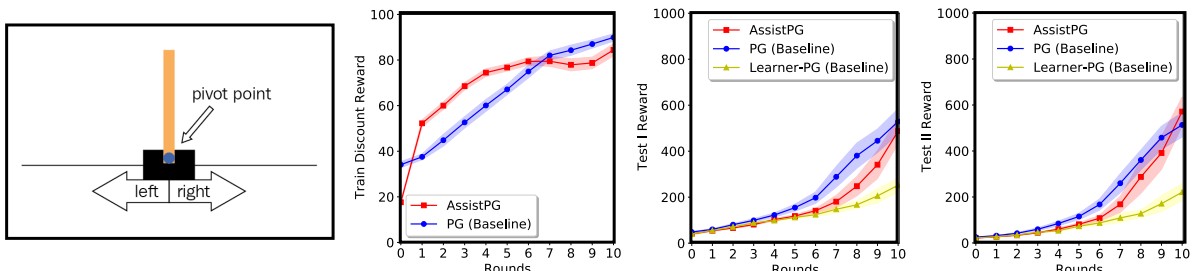

Figure 9: Comparison of AssistPG, PG, and Learner-PG in the CartPole game.

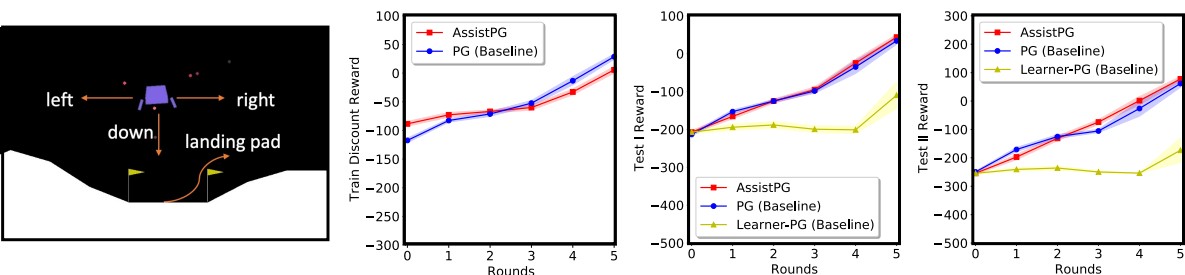

Figure 10: Comparison of AssistPG, PG, and Learner-PG in the LunarLander game.

We assume that the learner and the provider can query episode data by interacting with diverse environments. Specifically, for the CartPole problem, we parameterize the environment using the pole length. Both the learner and the provider train their control policies by playing 5 Cartpole games with the pole length randomly generated from Uniform$(4, 5)$ (for the learner) and Uniform$(0, 1)$ (for the provider). For the LunarLander problem, we parameterize the environment using the engine power. Both the learner and provider train their control policies by playing 10 LunarLander games with the engine power randomly generated from Uniform$(10, 15)$ (for the learner) and Uniform$(35, 40)$ (for the provider). Moreover, we consider two sets of testing environments that are uniform ("Test I") and non-uniform ("Test II"), respectively. For the CartPole problem, Test I environments randomly generate the pole length from Uniform$(0, 5)$, and Test II environments randomly generate the pole length from Beta$(1, 5)$ with probability 0.2 and Uniform$(0, 5)$ with probability 0.8. For the LunarLander problem, Test I environments randomly generate the engine power from Uniform$(10, 40)$, and Test II environments randomly generate the engine power as $30r + 10$, where $r \sim$ Beta$(5, 1)$.

We test AssistPG on both RL problems and compare its performance with two baselines: the standard PG (using centralized episode data) and the Learner-PG (using only the learner's episode data). All these algorithms are implemented with the learning rate $5 \times 10^{-3}$ and episode batch size 32. We model the policy using a three-layer feed-forward neural network with 4 and 32 hidden neurons for CartPole and LunarLander, respectively. Moreover, for AssistPG, we fix the total number of assistance rounds to be 10 and 5 for CartPole and LunarLander, respectively. The total number of local PG iterations in each assistance round is fixed to be 20 for both problems. We also set the sampling period to be four, namely, the learner and the provider record their local model and discounted training reward for every four local PG iterations. Fig. 9 and 10 plot the discounted training rewards (collected in local environments only), Test I cumulative rewards, and Test II cumulative rewards against the assistance round obtained by all these algorithms, for solving the CartPole and LunarLander problems, respectively. Here, one assistant round is interpreted as 20 local PG iterations for algorithms other than AssistPG.

Fig. 9 indicates that AssistPG outperforms Learner-PG, when the testing environments include diverse lengths of poles. Also, AssistPG can achieve comparable performance to that of the PG with centralized data. Moreover, Fig. 10 indicates that AssistPG can swiftly adapt to scenarios out of their comfort zone (namely the training environments) in only a few rounds. These experiments demonstrate that our assisted learning framework can help the learner significantly improve the quality of the policy for handling complex RL problems.

## 4.4  Visualization of LunarLander Experiment

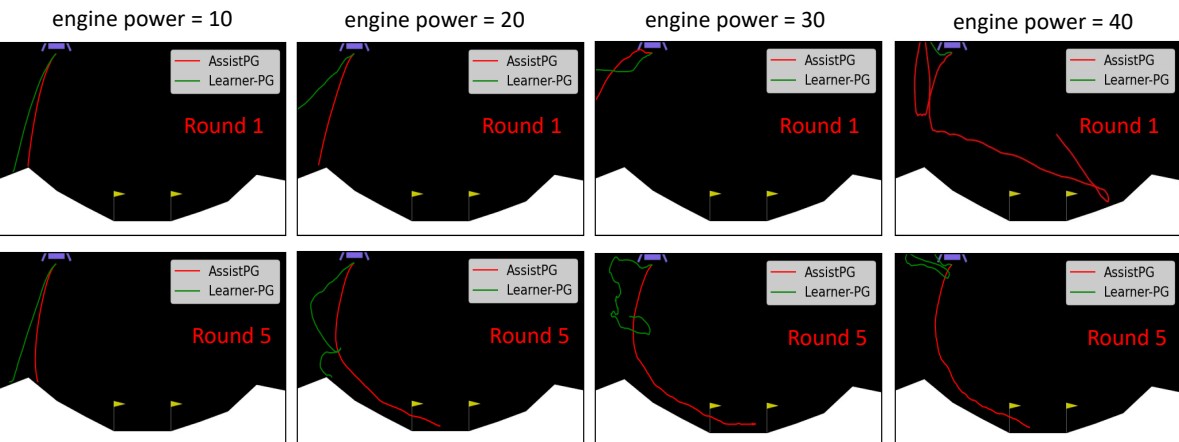

Figure 11: Landing traces of LunarLander with engine power = 10, 20, 30, 40 trained by AssistPG and Learner-PG.

In this section, we visualize the landing trace of the LunarLander trained by AssistPG and Leaner-PG in different test environments.

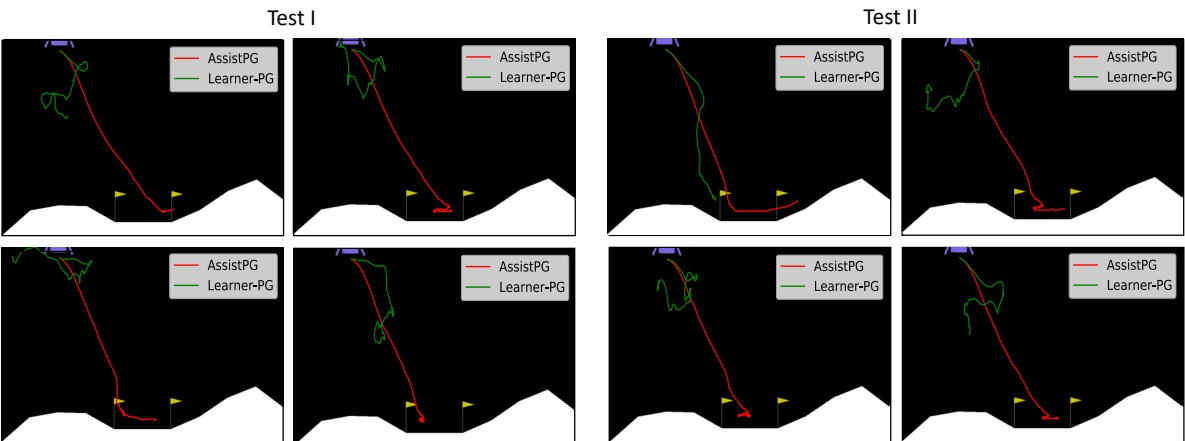

Figure 12: Landing traces with engine power $\sim \mathrm{Uniform}(10, 40)$ and $\sim 30 \cdot \mathrm{Beta}(5, 1) + 10$ trained by AssistPG and Learner-PG.

Specifically, we consider a fixed map and set the engine power of the lander to be 10, 20, 30, and 40, respectively. In each setting, we train the lander using both AssistPG and Learner-PG for $R = 5$ rounds. After each round of training, we let the lander play an episode using the trained model and plot the corresponding landing trace. These traces are plotted in Fig. 11. From these figures, it can be seen that the lander with engine power 20-40 trained by AssistPG can successfully land on the landpad after 5 rounds of assisted learning. As a comparison, the lander trained by Learner-PG cannot even land after 5 rounds of training. This demonstrates the advantage of AssistPG. On the other hand, when the lander has a small engine power 10, it is challenging for both algorithms to land the lander properly, as the engine cannot provide sufficient acceleration. Moreover, after 5 rounds of training (using both AssistPG and Learner-PG), we test the lander in both the test environment I ("Test I") and II ("Test II"), and plot the landing traces in Fig. 12. Here, for each test, we consider a fixed map and randomly generate 4 different engine powers from $\mathrm{Uniform}(10, 40)$ (for Test I) and $30 \cdot \mathrm{Beta}(5, 1) + 10$ (for Test II).

From both figures, it can be seen that the lander trained by the AssistPG lands more smoothly in all test environments under diverse engine powers than that trained by the Learner-PG. The video version for the CartPole and LunarLander games can be accessed from the anonymous link `https://www.dropbox.com/sh/oz2jswj36li4lkh/AADaQn4Nj67v9mdIHKDLN6nAa?dl=0`. In the CartPole game, four videos record the performance of AssistPG and Learner-PG against the first five rounds with pole lengths equaling 1, 2, 3, and 4, respectively. Another two videos record 10 plays in the test environment I and II, respectively. In all the plays, both AssistPG and Learner-PG use the model trained from the fifth round. In the LunarLander game, four videos record the performance of AssistPG and Learner-PG against the first five rounds with engine power equaling 10, 20, 30, and 40, respectively. Another two videos record 10 plays in the test environment I and II, respectively. In all the plays, both AssistPG and Learner-PG use the model trained from the fifth round. The videos show that with assistance from the provider, the learner can quickly generalize its model to more diverse environments.

## 5   Conclusion

This work develops a learning framework for assisting organizational learners to improve their learning performance with limited imbalanced data. In particular, the proposed AssistDeep and AssistPG allow the provider to assist the learner's training process and significantly improve its model quality within only a few assistance rounds. We demonstrate the effectiveness of both assisted learning algorithms through experimental studies. In the future, we expect that this learning framework can be integrated with other learning frameworks such as meta-learning and multi-task learning. A limitation of this study is that it only considers a pair of learner and provider. An interesting future direction is to emulate the current assisted learning framework to allow multiple learners or service providers.

**Acknowledgments**

The work of Cheng Chen and Yi Zhou was supported in part by U.S. National Science Foundation under the Grant. Nos. CCF-2106216, DMS-2134223 and CAREER-2237830. The work of Jiaying Zhou was supported in part by Army Research Office under grant number W911NF-20-1-0222. The work of Jie Ding was supported in part by U.S. National Science Foundation under the Grant. Nos. ECCS-2038603, DMS-2134148, and CNS-2220286.

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

# A Appendix

## A.1 Federated Learning (FL)

Federated learning is an emerging distributed learning framework (Shokri & Shmatikov, 2015; Konecny et al., 2016; McMahan et al., 2017; Zhao et al., 2018; Li et al., 2020; Diao et al., 2021; 2022) that aims to learn a global model using the average of local models trained by numerous smart devices with possibly imbalanced/heterogeneous data. The existing federated learning algorithms require frequent transmissions of local model parameters. This is different from our solution designed for the organizational learning scenarios, where each learner is an organization that often has unconstrained communication and computation resources, but is restricted to interacting with external service providers. Our solution aims to help the learner improve learning performance within ten rounds, while federated learning needs many more rounds.

### A.1.1 Comparison between FedAvg and AssistDeep

**Experimental Setup.** In this subsection, we compare the standard FedAvg algorithm (McMahan et al., 2017)for federated learning with centralized SGD (baseline), Learner-SGD (baseline) and our AssistDeep algorithm. We use the same datasets (CIFAR-10 and SVHN) and models (AlexNet and ResNet-18) as in Section 4.2 and keep all settings and hyperparameters unchanged. We implement all algorithms using the SGD optimizer with a learning rate of 0.01 for AlexNet and a learning rate of 0.1 for ResNet-18. The provider's data are uniformly sampled from all classes and are therefore balanced. The learner's data are also sampled with the same size as the provider's, but are taken from a single class and are therefore extremely imbalanced. For FedAvg, we treat the learner and provider as two federated learning clients/agents and use the same local data and local SGD iteration budgets as AssistDeep.

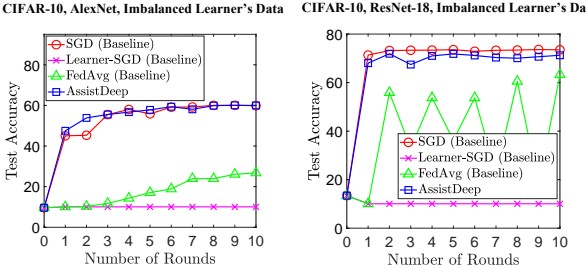

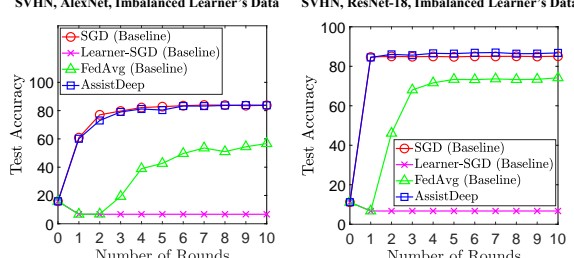

Figure 13: Comparison of AssistDeep, FedAvg, SGD, and Learner-SGD with imbalanced learner's data in training an AlexNet (left) and ResNet-18 (right) on CIFAR-10.

Figure 14: Comparison of AssistDeep, FedAvg, SGD, and Learner-SGD with imbalanced learner's data in training an AlexNet (left) and ResNet-18 (right) on SVHN.

**Comparison Results for CIFAR-10 Dataset.** We first compare these algorithms with extremely imbalanced learner's data in training an AlexNet and ResNet-18 on the CIFAR-10 dataset, respectively. In Fig. 13, we plot the test accuracy against the number of assistance rounds for AlexNet (left) and ResNet-18 (right), respectively. For SGD, Learner-SGD and AssistDeep, almost the same results can be observed as those shown in Fig. 3 (top left) and the same conclusions can be drawn as those made in subsection 4.2.1 (when the learner has extremely imbalanced data). Additionally, it can be observed that our AssistDeep algorithm converges much faster and achieves a much better test performance than FedAvg for both AlexNet and ResNet-18. Specifically, with AlexNet, our AssistDeep achieves about 60% test accuracy while FedAvg achieves about 27% accuracy; with ResNet-18, AssistDeep achieves about 71% test accuracy while FedAvg achieves about 40% (averaged) accuracy. This is because FedAvg simply averages the last local models generated by the two clients, and one of them is trained on highly imbalanced learner's data, which causes a significant bias. These results demonstrate that our proposed AssistDeep algorithm is more robust to data imbalance compared to standard federated learning.

**Comparison Results for SVHN Dataset.** In Fig. 14, we repeat the above experiments with the SVHN dataset using AlexNet (left) and ResNet-18 (right), respectively. From these results, one can make the

same observations as the previous CIFAR-10 results, which show that our proposed AssistDeep significantly outperforms FedAvg on diverse types of datasets. Specifically, with AlexNet, our AssistDeep achieves about 84% test accuracy while FedAvg achieves about 57% accuracy; with ResNet-18, AssistDeep achieves about 87% test accuracy while FedAvg achieves about 74% accuracy.

**Remark 1.** Based on the experimental settings above, we can quantify the communication load of both AssistDeep and FedAvg as follows. In AssistDeep, as we described in Algorithm 1 and Section 4.2, the number of learning rounds $R = 10$, the sampling period $I = 50$ iterations, and the numbers of local training iterations for the learner and provider are $T$ and $T'$ respectively, where $T + T' = 2000$. Suppose the model consists of $n$ parameters, then the total communication load is $nR(T + T')/I = 400n$ for AssistDeep to converge. Note that AssistDeep also communicates the losses between the learner and provider, but it occupies little communication so we just neglect them. As a comparison, the FedAvg algorithm in federated learning transmits two models per round but usually requires hundreds of rounds to converge, so its total communication load is of a similar scale to that of AssistDeep. This proves that our assisted learning algorithm does not strictly rely on unrestricted/unlimited communication resources.

### A.2 AssistKD: Assisted Learning with Knowledge Distillation

### A.2.1 Knowledge Distillation

Knowledge distillation (KD) is a machine learning technique that trains one classifier, called the student, using the outputs (also known as soft labels, which is a vector of probability scores assigned to each class) of another classifier, called the teacher (Hinton et al., 2015). It has been found to be often more efficient to train classifiers using soft labels rather than ground truth labels. The basic idea behind KD is to train a student model to match the output logits of a teacher model.

When defining the logit label of the teacher's model, a higher temperature $t$ represents the softer distribution of output classes. Suppose the output has $c$ classes. The $i$-th entry in the teacher's logit label is defined as

$$q_{T,i}(\mathbf{x}, t) = \frac{\exp(z_{T,i}(\mathbf{x})/t)}{\sum_{i=1}^{c} \exp(z_{T,i}(\mathbf{x})/t)},$$

where $z_{T,i}$ represents the pre-softmax layer associated with the $i$-th class. Similarly, we define the $i$-th entry in the student's logit label as

$$q_{S,i}(\mathbf{x}, \beta, t) = \frac{\exp(z_{S,i}(\mathbf{x}, \beta)/t)}{\sum_{i=1}^{c} \exp(z_{S,i}(\mathbf{x}, \beta)/t)},$$

where $\beta$ is the training parameters in student's model, and $z_{S,i}(\mathbf{x}, \beta)$ is the pre-softmax layer associated with the $i$-th class given training model parameters $\beta$. We use the following loss function (Hinton et al., 2015) for knowledge distillation:

$$\mathfrak{L}(y, \mathbf{x}, \beta) = -\alpha \sum_{i=1}^{c} q_{T,i}(\mathbf{x}, t) \log q_{S,i}(\mathbf{x}, \beta, t) - (1 - \alpha) \sum_{i=1}^{c} \mathbf{1}(y = i) \log q_{S,i}(\mathbf{x}, \beta, 1), \tag{3}$$

where $\alpha$ is a hyper-parameter that decides the weighting of soft labels and hard labels, and $y$ is the ground truth label. Here, the first term in the above loss corresponds to the distillation loss between the teacher's soft label and the student's soft label, while the second term corresponds to the student's cross-entropy loss.

We adapt KD to the assisted learning framework and call it AssistKD. In AssistKD, knowledge distillation occurs in two stages: 1) When a learner learns its local models, it will distill every model generated in the local trajectory of models to the shared model architecture. To save computation, we use each distilled model as a warm start for distilling the next in the trajectory. 2) When a learner picks the model that yields the lowest global loss, the learner will distill the chosen model to its local model architecture to initialize the training of its local models

### A.2.2 Experiments

We conduct an experiment where the learner and the provider train a ResNet-18 as their local training model. They communicate with each other by transmitting a smaller-scale CNN model (shared between learner and

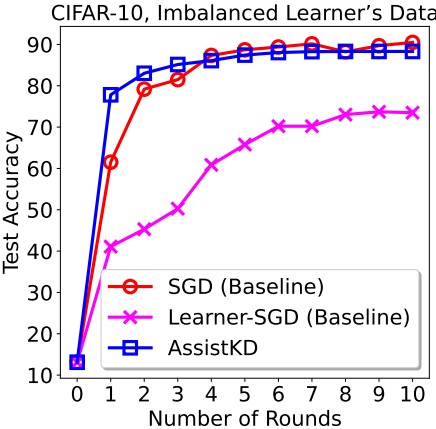

Figure 15: Comparison of AssistKD (CNN as the communication model), SGD (using model ResNet-18), and Learner-SGD (using model ResNet-18) on CIFAR-10.

provider). In this example, the shared model consists of six convolutional layers and three fully connected layers. It only has half of the trainable parameters compared with the trainable parameters in ResNet-18.

We test the performance of AssistKD on CIFAR-10 by comparing it with two baselines: SGD (using centralized data $\mathcal{D}^{(\mathrm{L,P})}$) and Learner-SGD (using only the learner's data $\mathcal{D}^{(\mathrm{L})}$). We implement all these algorithms using the SGD optimizer with a learning rate of 0.1 and batch size of 256 to train CNN and ResNet-18 on CIFAR-10. We assign different SGD iterations to the aforementioned two-stage knowledge distillation. In particular, we let the SGD iterations in the first stage be 10. In the second stage, we choose the SGD iterations that traverse the local training data for 60 times, i.e., 60 epochs. We choose hyperparameter $t = 1$ and $\alpha = 0.9$ in (3).

We compared these algorithms with imbalanced learner data. In particular, the provider's data are uniformly sampled from all classes and are therefore balanced. The size of the learner's data is only 20% of the provider's sample size. Moreover, half of the learner data is taken from a single class, and the other half is uniformly sampled from other classes. We fix the number of assistance rounds to be 10. The total number of local SGD iterations in each assistance round is fixed to be 2000. We assign the SGD iteration budget to the learner and provider in proportion to their data samples' sizes. Both the learner and provider record their local training models and local loss values for every $I = 50$ SGD iteration, which is the sampling period.

The test performance is shown in Fig. 15. From the plot, the performance of AssistKD approaches that of the standard SGD which trains a larger model (ResNet-18) on the centralized data. Moreover, the AssistKD significantly outperforms the Learner-SGD which trains on the learner's data alone. This demonstrates that our AssistKD enables the learner and provider to transmit models of different sizes/scales while keeping a similar learning performance to the original centralized learning.

## A.3 Proof of Theorem 1

For brevity, we denote the loss functions $f(\cdot; \mathcal{D}^{(\mathrm{L})})$, $f(\cdot; \mathcal{D}^{(\mathrm{P})})$, $f(\cdot; \mathcal{D}^{(\mathrm{L,P})})$ as $f_{\mathrm{L}}, f_{\mathrm{P}}, f_{\mathrm{L,P}}$, respectively. We let $\|\cdot\|$ denote the Euclidean norm.

Consider any assistance round $r$. Recall that the learner L initializes with the output model obtained from the previous round, namely $\theta_0^{(\mathrm{L}),r} = \theta^{r-1}$. In the local training, learner L performs $T$ local gradient descent steps and generates the trajectory $\{\theta_t^{(\mathrm{L}),r}\}_{t=0}^T$. Then, provider P picks the best model from this trajectory that achieves the minimum global loss, and we denote this model as $\theta_k^{(\mathrm{L}),r}$ for certain $k \in \{0, ..., T\}$. It is clear that $f_{\mathrm{L,P}}(\theta_k^{(\mathrm{L}),r}) \leq f_{\mathrm{L,P}}(\theta_t^{(\mathrm{L}),r})$ for all $t$. By smoothness of the global loss and the gradient descent update

rule, we have that

$$f_{\mathrm{L,P}}(\theta_{k+1}^{(\mathrm{L}),r}) \le f_{\mathrm{L,P}}(\theta_k^{(\mathrm{L}),r}) + \langle \theta_{k+1}^{(\mathrm{L}),r} - \theta_k^{(\mathrm{L}),r}, \nabla f_{\mathrm{L,P}}(\theta_k^{(\mathrm{L}),r}) \rangle + \frac{L}{2}\|\theta_{k+1}^{(\mathrm{L}),r} - \theta_k^{(\mathrm{L}),r}\|^2$$

$$= f_{\mathrm{L,P}}(\theta_k^{(\mathrm{L}),r}) + \langle -\eta \nabla f_{\mathrm{L}}(\theta_k^{(\mathrm{L}),r}), \nabla f_{\mathrm{L,P}}(\theta_k^{(\mathrm{L}),r}) \rangle + \frac{L\eta^2}{2}\|\nabla f_{\mathrm{L}}(\theta_k^{(\mathrm{L}),r})\|^2.$$

Since $f_{\mathrm{L,P}}(\theta_k^{(\mathrm{L}),r}) \le f_{\mathrm{L,P}}(\theta_{k+1}^{(\mathrm{L}),r})$, the above inequality further implies that

$$\langle \nabla f_{\mathrm{L}}(\theta_k^{(\mathrm{L}),r}), \nabla f_{\mathrm{L,P}}(\theta_k^{(\mathrm{L}),r}) \rangle \le \frac{L\eta}{2}\|\nabla f_{\mathrm{L}}(\theta_k^{(\mathrm{L}),r})\|^2 \le \frac{LG^2\eta}{2}. \tag{4}$$

Next, consider the local training of the provider P. Recall that provider P initializes with the best model sent by the learner, namely $\theta_0^{(\mathrm{P}),r} = \theta_k^{(\mathrm{L}),r}$. In the local training, the provider P performs $T'$ local gradient descent steps and generates the trajectory $\{\theta_t^{(\mathrm{P}),r}\}_{t=0}^{T'}$. According to the smoothness of the global loss, we have that

$$f_{\mathrm{L,P}}(\theta_1^{(\mathrm{P}),r}) \le f_{\mathrm{L,P}}(\theta_0^{(\mathrm{P}),r}) + \langle \theta_1^{(\mathrm{P}),r} - \theta_0^{(\mathrm{P}),r}, \nabla f_{\mathrm{L,P}}(\theta_0^{(\mathrm{P}),r}) \rangle + \frac{L}{2}\|\theta_1^{(\mathrm{P}),r} - \theta_0^{(\mathrm{P}),r}\|^2$$

$$= f_{\mathrm{L,P}}(\theta_0^{(\mathrm{P}),r}) + \langle -\eta \nabla f_{\mathrm{P}}(\theta_0^{(\mathrm{P}),r}), \nabla f_{\mathrm{L,P}}(\theta_0^{(\mathrm{P}),r}) \rangle + \frac{L\eta^2}{2}\|\nabla f_{\mathrm{P}}(\theta_0^{(\mathrm{P}),r})\|^2$$

$$= f_{\mathrm{L,P}}(\theta_0^{(\mathrm{P}),r}) + \langle -\eta \nabla f_{\mathrm{P}}(\theta_k^{(\mathrm{L}),r}), \nabla f_{\mathrm{L,P}}(\theta_k^{(\mathrm{L}),r}) \rangle + \frac{L\eta^2}{2}\|\nabla f_{\mathrm{P}}(\theta_0^{(\mathrm{P}),r})\|^2$$

$$= f_{\mathrm{L,P}}(\theta_0^{(\mathrm{P}),r}) - \eta\big(\|\nabla f_{\mathrm{L,P}}(\theta_k^{(\mathrm{L}),r})\|^2 - \langle \nabla f_{\mathrm{L}}(\theta_k^{(\mathrm{L}),r}), \nabla f_{\mathrm{L,P}}(\theta_k^{(\mathrm{L}),r}) \rangle \big) + \frac{L\eta^2}{2}\|\nabla f_{\mathrm{P}}(\theta_0^{(\mathrm{P}),r})\|^2$$

$$\le f_{\mathrm{L,P}}(\theta_0^{(\mathrm{P}),r}) - \eta\|\nabla f_{\mathrm{L,P}}(\theta_k^{(\mathrm{L}),r})\|^2 + \frac{LG^2\eta^2}{2} + \frac{LG^2\eta^2}{2}, \tag{5}$$

where the last inequality utilizes Inequality (4) and the boundedness of the gradient. Denote $\theta_k^{(\mathrm{P}),r}$ as the best model from this trajectory that achieves the minimum global loss. The above Inequality (5) and Proposition 1 further imply that

$$f_{\mathrm{L,P}}(\theta^r) = f_{\mathrm{L,P}}(\theta_{k'}^{(\mathrm{P}),r}) \quad (\text{for some } k' \in \{0,\dots,T'\})$$

$$\le f_{\mathrm{L,P}}(\theta_1^{(\mathrm{P}),r})$$

$$\le f_{\mathrm{L,P}}(\theta_0^{(\mathrm{P}),r}) - \eta\|\nabla f_{\mathrm{L,P}}(\theta_k^{(\mathrm{L}),r})\|^2 + LG^2\eta^2$$

$$\le f_{\mathrm{L,P}}(\theta^{r-1}) - \eta\|\nabla f_{\mathrm{L,P}}(\theta_k^{(\mathrm{L}),r})\|^2 + LG^2\eta^2$$

$$\le f_{\mathrm{L,P}}(\theta^{r-1}) - \eta\|\nabla f_{\mathrm{L,P}}(\theta_0^{(\mathrm{L}),r})\|^2 - \eta\|\nabla f_{\mathrm{L,P}}(\theta_k^{(\mathrm{L}),r}) - \nabla f_{\mathrm{L,P}}(\theta_0^{(\mathrm{L}),r})\|^2$$

$$\quad + 2\eta\|\nabla f_{\mathrm{L,P}}(\theta_k^{(\mathrm{L}),r}) - \nabla f_{\mathrm{L,P}}(\theta_0^{(\mathrm{L}),r})\| \cdot \|\nabla f_{\mathrm{L,P}}(\theta_0^{(\mathrm{L}),r})\| + LG^2\eta^2$$

$$\overset{(i)}{\le} f_{\mathrm{L,P}}(\theta^{r-1}) - \eta\|\nabla f_{\mathrm{L,P}}(\theta^{r-1})\|^2 + 3\eta^2 LTG^2,$$

where Inequality (i) uses the fact that

$$2\eta\|\nabla f_{\mathrm{L,P}}(\theta_k^{(\mathrm{L}),r}) - \nabla f_{\mathrm{L,P}}(\theta_0^{(\mathrm{L}),r})\|\|\nabla f_{\mathrm{L,P}}(\theta_0^{(\mathrm{L}),r})\| \le 2\eta G\|\nabla f_{\mathrm{L,P}}(\theta_k^{(\mathrm{L}),r}) - \nabla f_{\mathrm{L,P}}(\theta_0^{(\mathrm{L}),r})\|$$

$$\le 2\eta GL\|\theta_k^{(\mathrm{L}),r} - \theta_0^{(\mathrm{L}),r}\| = 2\eta^2 GL\left\|\sum_{j=0}^{k-1} \nabla f_{\mathrm{L}}(\theta_j^{(\mathrm{L}),r})\right\|$$

$$\le 2\eta^2 LTG^2.$$

Telescoping the above inequality over $r = 1,...,R$ and rearranging them, we obtain that

$$\frac{1}{R}\sum_{r=0}^{R-1} \|\nabla f_{\mathrm{L,P}}(\theta^r)\|^2 \le \frac{f_{\mathrm{L,P}}(\theta^0) - \inf_\theta f_{\mathrm{L,P}}(\theta)}{\eta R} + 3\eta LTG^2.$$

The result follows by choosing the step size

$$\eta = \sqrt{(f_{L,P}(\theta^0) - \inf_\theta f_{L,P}(\theta))/3RLTG^2}$$

and noting that

$$\min_{0 \le r \le R-1} \|\nabla f_{L,P}(\theta^r)\|^2 \le \frac{1}{R} \sum_{r=0}^{R-1} \|\nabla f_{L,P}(\theta^r)\|^2.$$

**Remark 2.** If the learning rate $\eta$ is small, Inequality (4) shows that the gradient $\nabla f_L(\theta_k^{(L),r})$ should not be well aligned with the gradient of the global loss $\nabla f_{L,P}(\theta_k^{(L),r})$. Intuitively, this is because $\theta_k^{(L),r}$ is the best model chosen from the training trajectory $\{\theta_t^{(L),r}\}_t$ that achieves the minimum global loss, and therefore the subsequent gradient update $\nabla f_L(\theta_k^{(L),r})$ must be badly correlated with $\nabla f_{L,P}(\theta_k^{(L),r})$ so that the global loss would increase in the next iteration.

### A.4 Assisted Learning with Differential Privacy (DP)

In recent years, data privacy has been formulated under different frameworks. A general direction is database privacy, such as the differential privacy (Dwork & Nissim, 2004), where private data are collected and managed by a trusted third party. Another general direction is local data privacy, to obfuscate data that have to be shared with potential adversaries. Examples include the local differential privacy (Evfimievski et al., 2003; Kasiviswanathan et al., 2011) that randomizes the raw data and the interval privacy (Ding & Ding, 2020) that reports random intervals containing the raw data. We note that a privacy framework usually concerns the protection of raw data or their identities, which is conceptually different from a learning framework. The integration of data privacy and learning has been recently studied. For example, (Abadi et al., 2016) developed a differentially-private SGD for deep learning, where the main idea is to perturb the gradient at each minibatch step.

**Experimental Setup.** We demonstrate that our AssistDeep can be implemented in the differential privacy (DP) framework to enhance the information privacy of both the learner and provider. We replace the local SGD training of both the learner and provider with the differentially private SGD proposed in (Abadi et al., 2016). We consider the $(\epsilon, \delta)$-DP with $\delta = 10^{-5}$ and $\epsilon = 1, 5, 10$ at each SGD step. The noise added to the gradients follow the Gaussian distribution $\mathcal{N}(0, 2\epsilon^{-2} \log \delta^{-1})$. We implement this differentially private AssistDeep with imbalanced and limited learner's data ($\gamma_L = 0.3$), and all other experimental hyperparameters and settings remain the same as Section 4.2. Using the strong composition formula (Kasiviswanathan et al., 2011), the privacy after $T$ SGD steps and batch fraction $q$ satisfies $(\epsilon', \delta')$-DP, with $\epsilon' = 2q\epsilon\sqrt{T \log(1/\delta)}$, $\delta' = qT\delta$. We calculated that the learner satisfies $(15.5, 0.001)$-DP, and the provider satisfies $(5.2, 0.001)$-DP (after rounding).

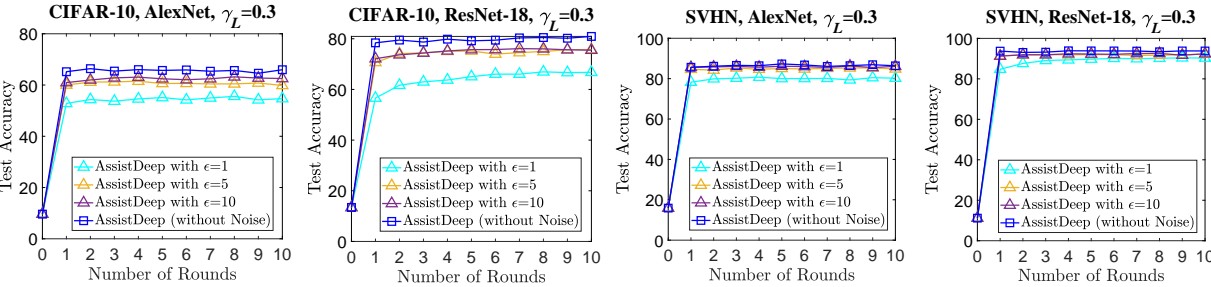

Figure 16: Comparison of original AssistDeep (without noise) and AssistDeep algorithms with different levels of DP noises ($\epsilon = 1, 5, 10$) under $\gamma_L = 0.3$.

**Comparison Results.** Fig. 16 shows the results of training both AlexNet and ResNet-18 on CIFAR-10 and SVHN datasets, respectively. We can see that for a more restricted privacy level (namely small $\epsilon$), more noise

is added and the performance of AssistDeep degrades. However, when the $\epsilon$ is large enough, the performance of differentially private AssistDeep is very similar to that of the original AssistDeep. Therefore, the results demonstrate that our AssistDeep can achieve reasonable levels of differential privacy without significantly degrading the performance.

## A.5 Effect of Checkpoints Transferred from Learner to Provider

**Experimental Setup.** To evaluate the importance of the checkpoints passed from the learner to the provider in our algorithm design, we modified the original AssistDeep algorithm to let the provider train on its local data using weights initialized from the model generated in the learner's last local training iteration (instead of the checkpoint model that achieves the minimum global loss). We name this algorithm as Revised-AssistDeep. Then under imbalanced and limited learner's data ($\gamma_L = 0.3$) and different numbers of learner's local iterations ($T = 2000, 6000, 12000$), we conducted experiments to train the AlexNet on CIFAR-10 dataset for both AssistDeep and Revised-AssistDeep to compare their performance. And all other experimental hyperparameters and settings remain the same as Section 4.2.

**Comparison Results.** For each experiment, we averaged the test accuracies of the last four rounds to approximate the model's fully trained performance, and put the results into Table 1. From all the results with different $T$ values (2000, 6000, 12000), it can be seen that Revised-AssistDeep performs worse than our AssistDeep. Specifically, under $T = 2000$, averaged test accuracies of AssistDeep and Revised-AssistDeep are 66.0% and 65.3% respectively; under $T = 6000$, averaged test accuracies of AssistDeep and Revised-AssistDeep are 65.6% and 64.8% respectively; under $T = 12000$, averaged test accuracies of AssistDeep and Revised-AssistDeep are 66.2% and 64.1% respectively. These results demonstrate that in the algorithm design of our AssistDeep, it is necessary for the provider to initialize its weights from the checkpoints passed by the learner to do the local training.

Table 1: Comparison of AssistDeep and Revised-AssistDeep in training an AlexNet on CIFAR-10

| Test Accuracy | | | |
|---|---|---|---|
| **Algorithm Name** | $T = 2000$ | $T = 6000$ | $T = 12000$ |
| AssistDeep | **66.0%** | **65.6%** | **66.2%** |
| Revised-AssistDeep | 65.3% | 64.8% | 64.1% |

## A.6 Provider-SGD: Centralized Model Trained on the Provider's Data Only

**Experimental Setup.** To compare the performance of our AssistDeep to that of the same model trained on the provider's data only (named Provider-SGD), we conduct experiments on Provider-SGD by training AlexNet on CIFAR-10 and SVHN datasets, respectively. Then we compare the performance of Provider-SGD with that of SGD, Learner-SGD and our AssistDeep under both imbalanced ($\gamma_L = 0.5$) and balanced ($\gamma_L = 1$) learner's data. All other experimental hyperparameters and settings remain the same as Section 4.2.

Table 2: Comparison of SGD, Learner-SGD, Provider-SGD, and AssistDeep in training an AlexNet on CIFAR-10 and SVHN

| Test Accuracy | | | | |
|---|---|---|---|---|
| **Algorithm Name** | **CIFAR-10, $\gamma_L = 0.5$** | **CIFAR-10, $\gamma_L = 1$** | **SVHN, $\gamma_L = 0.5$** | **SVHN, $\gamma_L = 1$** |
| SGD | **66.6%** | **68.1%** | **86.8%** | **87.5%** |
| Learner-SGD | 58.4% | 63.2% | 82.4% | 83.9% |
| Provider-SGD | 64.0% | 64.0% | 85.6% | 85.6% |
| AssistDeep | 66.0% | **68.1%** | **86.8%** | 87.3% |

**Comparison Results.** For each experiment, we averaged the test accuracies of the last nine rounds to approximate the model's fully trained performance, and put the results into Table 2. Both the CIFAR-10 and

SVHN results show that Provider-SGD performs worse than AssistDeep and SGD, but better than Learner-SGD. Specifically, for CIFAR-10, Provider-SGD achieves about 64.0% test accuracy while our AssistDeep achieves about 66.0% accuracy under imbalanced learner's data and about 68.1% accuracy under balanced learner's data; for SVHN, Provider-SGD achieves about 85.6% test accuracy while our AssistDeep achieves about 86.8% accuracy under imbalanced learner's data and about 87.3% accuracy under balanced learner's data. Thus from all the results, one can see that our AssistDeep performs better than Provider-SGD. This proves that under both imbalanced and balanced learner's data, our AssistDeep outperforms Provider-SGD since the former can leverage both the learner's data and provider's data to improve the model's generalization performance while the latter leverages only the provider's data.

### A.7 Resampled-Learner-SGD: Learner-SGD with Balanced Learner's Data

To address the issue of the learner's data imbalance, we resample the data so that each batch will sample data points from each class with equal probability. Specifically, for the learner's local data, we copy the samples of each minor class to augment them and align them with the samples of the major class. In this case, the learner has the data balance. Then we implemented the Learner-SGD with such a resampling strategy and named it Resampled-Learner-SGD.

**Experimental Setup.** Under imbalanced and limited learner's data ($\gamma_L = 0.3$), we compare the Resampled-Learner-SGD with the original Learner-SGD and our AssistDeep in training AlexNet and ResNet-18 on CIFAR-10 and SVHN datasets, respectively. And all other experimental hyperparameters and settings remain the same as Section 4.2.

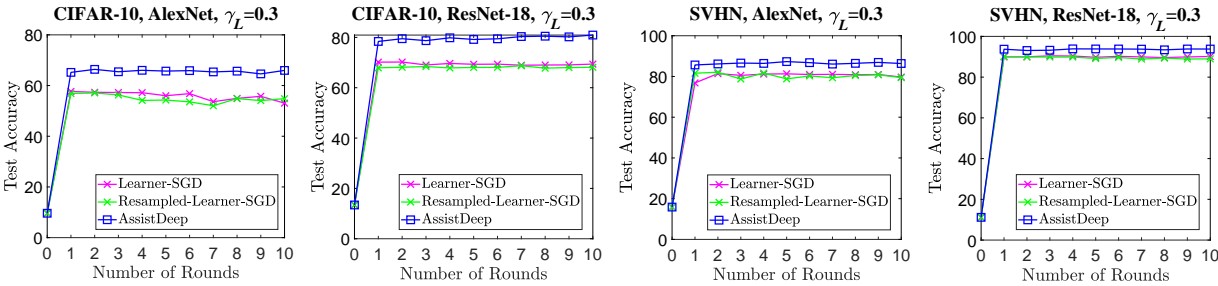

Figure 17: Comparison of original Learner-SGD, Resampled-Learner-SGD, and AssistDeep under $\gamma_L = 0.3$.

**Comparison Results.** From all the results shown in Fig. 17, it can be seen that the performance of Resampled-Learner-SGD is very similar to that of Learner-SGD but worse than AssistDeep, which proves the advantage of our AssistDeep over the Resampled-Learner-SGD regardless of models and training data.

### A.8 Effect of Number of Local Optimization Iterations

To explore the effect of the total number of local optimization iterations in the training process, we compare the performance of AssistDeep under the different total numbers of local iterations in each assistance round. Note that as we described in Section 4.2, we assign the iteration budget of the total number of local optimization iterations to the learner ($T$ iterations) and provider ($T'$ iterations) in proportion to their local data sizes. Also note that in our paper, the total number of iterations is fixed as 2000 by default, namely $T + T' = 2000$.

**Experimental Setup.** Specifically, under imbalanced and limited learner's data ($\gamma_L = 0.3$), we compare AssistDeep's performance under several different values of the total number of iterations, namely $T + T' = 500, 1000, 2000, 4000$. We conduct those experiments in training AlexNet and ResNet-18 on CIFAR-10 and SVHN datasets, respectively. And all other experimental hyperparameters and settings remain the same as Section 4.2.

**Comparison Results.** From all the results shown in Fig. 18, one can see that under different values of the total number of local optimization iterations (i.e., $T + T'$), AssistDeep achieves very similar performance, indicating that our AssistDeep is not sensitive to this hyper-parameter. Regardless of models and training

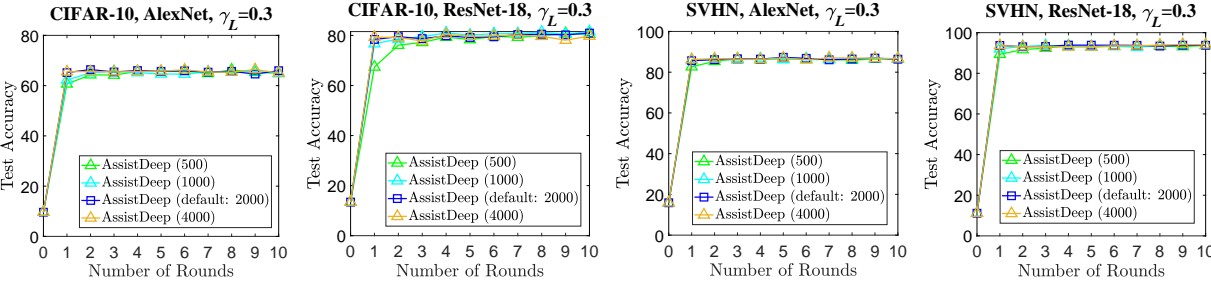

Figure 18: Comparison of AssistDeep algorithms with different numbers of local optimization iterations (500, 1000, 2000, 4000) under $\gamma_L = 0.3$.

data, the performance of AssistDeep almost keeps unchanged under the values of the total number of local iterations greater or equal to 2000, which is the reason that we set this hyper-parameter as 2000 in our paper.

### A.9 Explore the Transferability and Efficiency of AssistDeep When the Learner and Provider Hold Data from Different Categories or Domains

Most of our experiments assumed that learner L and provider P hold data from the same or very similar category and domain. In the real world, however, sometimes it is hard for the learner and provider to procure data from the same category and domain. Hence, it is necessary to conduct some experiments to prove the transferability and efficiency of our proposed Assisted Learning when the learner and provider have data from different categories or domains. To implement this, we conduct two groups of experiments: 1. learner's and provider's data come from different categories (but with the same domains), which means the major class data of learner's data does not appear in the provider's data; 2. learner's and provider's data come from different domains (but with the same categories), which means learner's and provider's data come from two different datasets. The details are presented as follows.

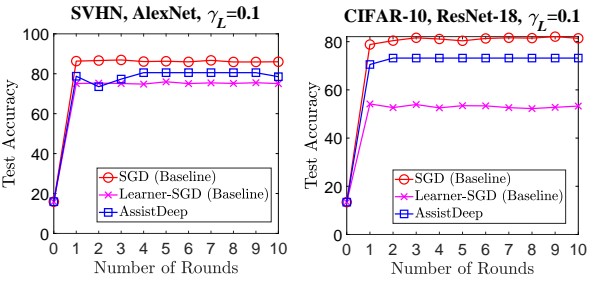

Figure 19: Comparison of AssistDeep, SGD, and Learner-SGD under imbalanced learner's data in training an AlexNet on SVHN (left) and a ResNet-18 on CIFAR-10 (right) when learner L and provider P hold data from different categories.

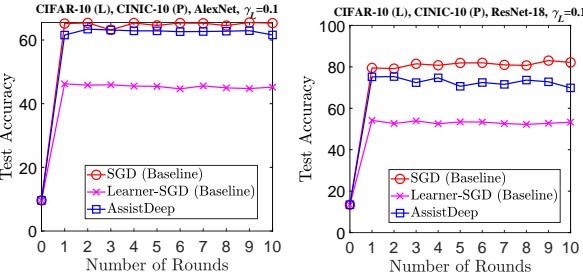

Figure 20: Comparison of AssistDeep, SGD, and Learner-SGD under imbalanced learner's data in training an AlexNet and a ResNet-18 respectively when learner L holds data from CIFAR-10 and provider P holds data from CINIC-10.

#### A.9.1 Group 1: Learner and Provider Hold Data with Different Categories

**Experimental Setup.** From SVHN and CIFAR-10 datasets respectively, suppose that we use the original data sampling method for both the learner and provider to get the data, but let the provider remove the data from the learner's major class. And all other settings and hyper-parameters keep the same as those described in Section 4.2. Then by training an AlexNet on SVHN and a ResNet-18 on CIFAR-10 respectively, we conduct experiments to compare the performance of our AssistDeep with that of SGD and Learner-SGD under limited imbalanced learner's data ($\gamma_L = 0.1$).

**Comparison Results.** From the results shown in Fig. 19, it can be seen that the performance of AssistDeep is much better than Learner-SGD and a little worse than SGD, which gives the same conclusion as that in Section 4.2. The results prove that our proposed Assisted Learning algorithm (AssistDeep) can still work efficiently even if the learner's and provider's data come from different categories, which further proves the excellent transferability of our proposed algorithm for data with different categories.

### A.9.2 Group 2: Learner and Provider Hold Data from Different Domains

**Experimental Setup.** With the original data sampling method, suppose that the learner samples data from the CIFAR-10 dataset and the provider samples data from the CINIC-10 dataset (Darlow et al., 2018), while all other settings and hyper-parameters keep the same as those described in Section 4.2. Note that in this case, we evaluate and get the test performance on the learner's test data only. Then by training an AlexNet and a ResNet-18 respectively, we conduct experiments to compare the performance of our AssistDeep with that of SGD and Learner-SGD under limited imbalanced learner's data ($\gamma_L = 0.1$).

**Comparison Results.** From the results shown in Fig. 20, it can be seen that the performance of AssistDeep is much better than Learner-SGD and a little worse than SGD, which gives the same conclusion as that in Section 4.2. The results prove that our proposed Assisted Learning algorithm (AssistDeep) can still work efficiently even if the learner's and provider's data come from different domains, which further proves the excellent transferability of our proposed algorithm for data with different domains.

