# OpenReview forum: "Assisted Learning for Organizations with Limited Imbalanced Data"
_TMLR — Accepted by TMLR_

### Review · Reviewer_gXtW · 2022-12-27

**Summary Of Contributions:**

This paper studies the assisted learning problem with horizontal-splitting of data and proposes an algorithm called AssistDeep for the setup studied.  Assisted learning addresses the problem of a client with limited data trying to learn a high performing model with the assistance of a provider with access to more data subject to two key constraints: (1). both client and provider cannot share data and (2) learning must be completed within a small number of iterations due to resource constraints.  In the horizontal-splitting setting, the client and provider have data with the same features but the provider has a much more representative dataset than the client.  AssistDeep proceeds in three stages for each round of assisted learning:
1. The client trains a model on its local data and sends a subset of the model checkpoints and evaluation metrics observed during the learning process to the provider.
2. The provider evaluates the subset of checkpoints received on a combination of the client's data and its own data then selects the best one for further training.  Once training is complete, it sends back a subset of the checkpoints and evaluation metrics on provider's local data to the client.
3. The client selects the checkpoint from the provider that performs the best on the joint data.

Experiments on CIFAR10 and SVHN for the AssistDeep and CartPole and LunarLander for AssistPG show the proposed algorithms can achieve test performance comparable to an oracle trained on both client and provider data under different values of imbalance for the client data.

**Audience:**

Yes

**Broader Impact Concerns:**

None.

**Claims And Evidence:**

No

**Requested Changes:**

- Critical: Please investigate how the algorithm can be modified to use residuals so that privacy can be better preserved.
- Critical: Please provide theoretical analysis even in restricted settings for convergence guarantees.
- Critical: Please run additional experiments to answer the 3 questions above.
- Critical: Please discuss how sensitive DeepAssist is to T and T' and how you would go about tuning hyperparameters for the model on both client and provider.

**Strengths And Weaknesses:**

Strengths:
- The proposed method is simple and straightforward, making it easy to apply in practice.
- The writing is clear and well structured.

Weaknesses:
- The algorithm conflicts with the goal of preserving data privacy.  In particular, communicating multiple checkpoints to the provider and then back to the client can leak information about the local training data. In prior work on assisted learning, there is usually also the assumption that the model and training object stay private as well between client and provider (http://assisted-learning.org/) hence why Xian et al., 2020 communicated residuals instead of model weights.
- There is no theoretical analysis to accompany the proposed algorithm to either guarantee convergence or give bounds on convergence.
- The evaluation is insufficient.  There are a few questions that need to be answered:
  1. How important it is for the provider to train on it's local data using weights initialized with the checkpoints passed from the client?  How does the performance of AssistDeep compare to using a model just trained on the provider's data (e.g. ProviderSGD)?
  2. There also does not seem to be any adjustments to the client's training procedure to account for data imbalance (e.g. weighted loss). How do results look if this is incorporated for the LearnerSGD baseline?
  3. How do results depend on the relative size of the client and provider's data?
- AssistDeep requires specifying T and T' for the training duration of the client and provider models as well as standard hyperparameters for the model.  It is unclear how to tune hyperparameters in this setting.

---

> ### Author Response · Authors · 2023-02-14
> **Thank you for the time and comments. We address each comment below.**
>
> **Q1:**
>
> **Weaknesses:** The algorithm conflicts with the goal of preserving data privacy. In particular, communicating multiple checkpoints to the provider and then back to the client can leak information about the local training data. In prior work on assisted learning, there is usually also the assumption that the model and training object stay private as well between client and provider (http://assisted-learning.org/) hence why Xian et al., 2020 communicated residuals instead of model weights.
>
> **Requested Changes:** Please investigate how the algorithm can be modified to use residuals so that privacy can be better preserved.
>
> **A:** Thanks for the suggestion. We agree with the reviewer that data leakage is a fundamental problem in machine learning. We note that this work aims to develop assisted learning algorithms for agents with the same feature variables but different subjects, also called the "horizontal data splitting" scenario. The earlier work on assisted learning (Xian et al., 2020) mentioned by the reviewer studied an orthogonal scenario where agents hold different feature variables but for the same subjects (aligned according to a data ID), also called the "vertical data splitting" scenario. Therefore, the use of residuals in (Xian et al., 2020) is not applicable to our scenario, where the data observations of one agent are independent of those of the other agent (since they observe different subjects, unlike the scenario in Xian et al., 2020).
>
> To address the reviewer's concern on privacy, in the revised manuscript, we have considered an alternative approach to enhance privacy in the sense of differential privacy. In particular, we used the gradient perturbation approach that adds a suitable amount of additive noise to the stochastic gradients during model training so that it meets differential privacy under a pre-specified privacy budget. Specifically, with imbalanced and limited learner’s data ($\gamma_{L}=0.3$) and keeping all other settings the same as in Section 4.2, we conduct experiments with different models and training data to compare the performance of our AssistDeep algorithm under different levels of DP noises ($\epsilon=1,5,10$). The results are shown in Section A.4 of the Appendix in the revision. All the results show that our AssistDeep can achieve a reasonable differential privacy guarantee without significantly degrading the performance. This shows that injecting controlled noise into the local stochastic gradients has little impact on the overall performance of our assisted learning framework. We believe this is an interesting direction to explore in future work.
>
> **Q2:**
>
> **Weaknesses:** There is no theoretical analysis to accompany the proposed algorithm to either guarantee convergence or give bounds on convergence.
>
> **Requested Changes:** Please provide theoretical analysis even in restricted settings for convergence guarantees.
>
> **A:** Thanks for the suggestion. We have developed and added a theoretical convergence analysis of our AssistDeep to Section 2.3 in the revision. In our analysis, we show that our assisted learning algorithm provably converges to a stationary point in general smooth nonconvex optimization. The detailed proof of Theorem 1 is added to Section A.3 in the Appendix of the revision.
>
> **Q3:**
>
> **Weaknesses:** The evaluation is insufficient. There are a few questions that need to be answered:
>
> 1. (1) How important it is for the provider to train on its local data using weights initialized with the checkpoints passed from the client? (2) How does the performance of AssistDeep compare to using a model just trained on the provider's data (e.g. ProviderSGD)?
>
> 2. There also does not seem to be any adjustments to the client's training procedure to account for data imbalance (e.g. weighted loss). How do results look if this is incorporated for the LearnerSGD baseline?
>
> 3. How do results depend on the relative size of the client and provider's data?
>
> **Requested Changes:** Please run additional experiments to answer the 3 questions above.
>
> **A:** Thanks for the great suggestions. We have run all these required experiments and added the results to the revised paper to address the above three questions. Unless otherwise specified, these experiments adopted imbalanced and limited learner’s data (i.e., $\gamma_{L}=0.3$) and used the same settings as those in Section 4.2. We present the specific answers to the three questions as follows:
>
> **=== To Be Continued (more responses are appended in the following text field) ===**

---

### Review · Reviewer_h9EE · 2023-02-07

**Summary Of Contributions:**

This paper attempts to establish the framework of assisted learning in a context similar to horizontal federated learning, in which the feature space is shared but each participant has its own data.

**Audience:**

Yes

**Claims And Evidence:**

No

**Requested Changes:**

I hope the authors can address all the critical concerns mentioned above.

**Strengths And Weaknesses:**

### Pros:

This paper explores a new field of leveraging assisted learning for the scenario where multiple organizations hold their own data (including different subjects) but share the same feature space.

This setup is very similar to that of horizontal federated learning. While as mentioned in [1], which proposed (vertical) assisted learning, the difference between assisted learning and federated learning is that
- "Conceptually, **the objective of Federated Learning** is to exploit resources of massive edge devices **for achieving a global objective**. At the same time, **the general goal of Assisted Learning** is to facilitate multiple participants (possibly with rich resources) to autonomously **assist each other’s private learning tasks**."

[1] Xian, Xun, et al. "Assisted learning: A framework for multi-organization learning." Advances in neural information processing systems 33 (2020): 14580-14591.

### Cons:

However, I do have some concerns over the claims/assumptions/executions of this paper.

First and foremost, given that the crucial fact distinguishing assisted learning from federated learning lies in assisting each other's private learning tasks,  I am particularly concerned about the specific formulation of this paper that **sets the data hold by a single organization as imbalanced (be supposed to follow a private data distribution) but enforces the model of this organization to learn a global distribution via assisted learning**. This setup is in contradiction with the objective of assisting each organization's private learning tasks but actually follows the spirit of federated learning to achieve a global objective (each organization learns the global data distribution).

Second, this paper claims no data sharing across organizations (e.g., Section 1.1), but the proposed algorithm (e.g., Algorithm 1) requires the communication of a series of model parameters and losses, which can implicitly violate the no-data-sharing assumption. Thus it is indispensable to include a data-privacy guarantee here.

Last but not the least, this paper assumes that "In each assistance round, both the learner and the provider have access to sufficient computation and communication resources". In my view, the assumption of unrestricted commucations is unrealistic in practice because real-world applications probably have limited commucation bandwidths, which have motivated extensive studies in distributed learning. Therefore I expect this paper can quantify the necessary communications needed for horizontal assisted learning.

---

> ### Author Response · Authors · 2023-02-21
> **Thank you for the time and comments. We address each comment below.**
>
> **Q1:** I am particularly concerned about the specific formulation of this paper that **sets the data hold by a single organization as imbalanced but enforces the model of this organization to learn a global distribution via assisted learning**. This setup is in contradiction with the objective of assisting each organization's private learning tasks but actually follows the spirit of federated learning to achieve a global objective.
>
> **A:** Good point. To clarify, the assisted learning setup considered in this paper is a bit different, which aims to assist a learner by connecting it to an external service provider. Here, the assistive learning goal is **one-sided**, i.e., the provider promises to help the learner improve its performance on its own learning task, and the provider does not need the assistance service. This is a standard consulting model that is very useful in commercialized business.
>
> Regarding the global objective function of our assisted learning. Although it is the same as that of federated learning, our algorithm can be directly applied to more general formulations, e.g., the global objective function weights the loss on the provider's data by a parameter $\lambda \in (0,1)$ (based on the learner's preference).
>
> **Q2:** This paper claims no data sharing across organizations (e.g., Section 1.1), but the proposed algorithm (e.g., Algorithm 1) requires the communication of a series of model parameters and losses, which can implicitly violate the no-data-sharing assumption. Thus it is indispensable to include a data-privacy guarantee here.
>
> **A:**  Thanks for the great suggestion. We agree that data leakage is a fundamental problem in the whole field of machine learning. To address the reviewer's concern about data privacy, in the up-to-date manuscript, we have considered an approach to enhance data privacy in the sense of differential privacy (DP). In particular, we used the gradient perturbation approach that adds a suitable amount of additive noise to the stochastic gradients during model training so that it meets differential privacy under a pre-specified privacy budget. Specifically, with imbalanced and limited learner’s data ($\gamma_{L}=0.3$) and keeping all other settings the same as in Section 4.2, we conduct experiments with different models and training data to compare the performance of our AssistDeep algorithm under different levels of DP noises ($\epsilon=1,5,10$). The results are shown in Section A.4 of the Appendix in the revision. All the results show that our AssistDeep can achieve a reasonable differential privacy guarantee without significantly degrading the performance. This shows that injecting controlled noise into the local stochastic gradients has little impact on the overall performance of our assisted learning framework. We believe this is an interesting direction to explore in a future comprehensive work.
>
> **Q3:** This paper assumes that "In each assistance round, both the learner and the provider have access to sufficient computation and communication resources". In my view, the assumption of unrestricted communications is unrealistic in practice because real-world applications probably have limited communication bandwidths, which have motivated extensive studies in distributed learning. Therefore I expect this paper can quantify the necessary communications needed for horizontal assisted learning.
>
> **A:** We are sorry for the confusion. We want to clarify that our assisted learning algorithm does not strictly rely on unrestricted/unlimited communication. In our experiments, we just need **sufficient** communication (precisely, transmitting 40 local models) in each learning round.
>
> We further quantify the communication load of AssistDeep as follows. As we described in Algorithm 1 and Section 4.2, we run AssistDeep for $R$ learning rounds. In each learning round, the learner and provider run $T$ and $T'$ local training iterations, respectively. Also, the sampling period is $I$ iterations. Thus in each learning round, we know the learner and provider send $T/I$ and $T'/I$ local models, respectively. Suppose the model consists of $n$ parameters, then the total communication load of AssistDeep is $nR(T+T')/I$. In our experiments, given that $R = 10$, $I = 50$ and $T+T' = 2000$, our AssistDeep can converge with the total communication load $nR(T+T')/I = 400n$. As a comparison, the standard FedAvg algorithm in federated learning transmits two models per round (two agents with upload and download) but usually requires hundreds of rounds to converge, so its total communication load is of a similar scale to that of AssistDeep. We have added one remark to clarify this in Appendix A.1 of the revision.

---

### Review · Reviewer_emPx · 2023-02-26

**Summary Of Contributions:**

The article discusses the scenario where the provider holds much data and the learner has limited imbalanced data. The author propose ‘Assisted Learning Protocols’ that the learner can purchase assistance services (without data sharing) from the provider to enhance their model performance within a few rounds.

**Audience:**

Yes

**Claims And Evidence:**

Yes

**Requested Changes:**

1. The experimental setting, i.e., W1 in the above review.
2. Difference between assisted learning and federated learning should be compared.
3. More detailed and complex experiments (W3).

**Strengths And Weaknesses:**

Strengths:

1. The motivation is clear and the research question is well-defined.
2. The paper is well written and easy to follow.

Weakness:

1. My biggest concern is that the data used in this paper is in the ‘horizontal-splitting’ setting, which means the data in learner’s hand and provider’s hand is strictly i.i.d. However, I think it is very hard that the data is in the same domain and shares the same category. To prove the transferability of the proposed Assisted Learning, it is necessary to conduct experiments when the data in learner’s hand and provider’s hand is not from the same domain or has different categories.
2. Assisted Learning is very close to Federate Learning. Though the authors claim that Assisted Learning is within a few rounds. I would like to see Assisted Learning as a special case of Federate Learning. To show the necessity of Assisted Learning, ablation experiments of Federate Learning are needed. Specifically, how is the performance when Federate Learning methods are also conducted within a few rounds?
3. The experiments are not sufficient. The proposed method is very simple but experiments are only conducted on toy datasets(CIFA10) and outdated neural networks(AlexNet, ResNet18). More experiments on larger datasets(ImageNet) and modern neural networks(Vision Transformer) are encouraged.

---

> ### Author Response · Authors · 2023-03-12
> **Thank you for the time and comments. We address each comment below.**
>
> **W1:** My biggest concern is that the data used in this paper is in the ‘horizontal-splitting’ setting, which means the data in learner’s hand and provider’s hand is strictly i.i.d. However, I think it is very hard that the data is in the same domain and shares the same category. To prove the transferability of the proposed Assisted Learning, it is necessary to conduct experiments when the data in learner’s hand and provider’s hand is not from the same domain or has different categories.
>
> **A:** Thanks for the great suggestion, this is indeed an interesting setting. To test the transferability of our proposed Assisted Learning algorithm, we have conducted some additional experiments in which the learner and provider possess data from different domains or with different categories, as suggested by the reviewer. All experimental details and results are presented in Section A.9 in the Appendix of the revised paper.
>
> Specifically, we first explore the setting in which both the learner's and provider's data samples are from the same domain (e.g., CIFAR-10) but may belong to different classes. We consider a simplified setting in which the provider's data does not contain samples from the learner's major class. Here, the learner holds limited imbalanced data with a certain major class (i.e., $\gamma_{L}=0.1$), whereas the provider holds a large number of balanced data across different classes excluding the learner's major class. All other settings and hyper-parameters are the same as those described in Section 4.2. Then, we train an AlexNet on SVHN and a ResNet-18 on CIFAR-10, respectively, and compare the performance of our AssistDeep with that of SGD and Learner-SGD. It can be seen from Fig. 19 that the performance of AssistDeep is much better than Learner-SGD and slightly worse than SGD, which proves the effectiveness of our algorithm under this setting.
>
> We further explore the other setting in which learner's and provider's data come from different domains. Specifically, we consider that the learner holds data sampled from the CIFAR-10 dataset and the provider holds data sampled from the CINIC-10 [1] dataset. These two datasets share the same set of classes but their images are distinct from each other. Here, the learner's data is limited and imbalanced (i.e., $\gamma_{L}=0.1$), whereas the provider's data is big and balanced. All other settings and hyper-parameters are the same as those described in Section 4.2. We train an AlexNet and a ResNet-18, respectively, and compare the test performance of our AssistDeep with that of SGD and Learner-SGD on the learner's test data. It can be seen from Fig. 20 that the performance of AssistDeep is much better than Learner-SGD and slightly worse than SGD, which proves the transferability of our algorithm under this setting. As a future work, we believe our algorithm can be combined with domain adaptation techniques to further improve transferability.
>
> [1] L. N. Darlow, E. J. Crowley, A. Antoniou, and A. J. Storkey, "Cinic-10 is not imagenet or cifar-10," arXiv preprint arXiv:1810.03505, 2018.
>
> **W2:** Assisted Learning is very close to Federate Learning. Though the authors claim that Assisted Learning is within a few rounds. I would like to see Assisted Learning as a special case of Federate Learning. To show the necessity of Assisted Learning, ablation experiments of Federate Learning are needed. Specifically, how is the performance when Federate Learning methods are also conducted within a few rounds?
>
> **A:** Thank you for your great suggestion. We have conducted some ablation experiments to compare the standard Federated Learning method (i.e., FedAvg) with our AssistDeep method. Specifically, we train AlexNet and ResNet-18 on CIFAR-10 and SVHN, respectively, and consider highly imbalanced learner's data. The experimental details and results are presented in Appendix A.1 of the revision. From these results, it can be seen that FedAvg converges very slowly due to the highly imbalanced learner's data, whereas our AssistDeep has a much more robust performance. This is because FedAvg only linearly aggregates the models produced in the last iterations of local training, whereas AssistDeep performs screening over all the models produced in the local training and picks the best one. These ablation experiments demonstrate that AssistDeep performs significantly better than FedAvg when the learner's data is highly imbalanced.
>
> **=== To Be Continued (more responses are appended in the following text field) ===**

---

> > ### Author Response · Authors · 2023-03-12
> > **== Response Continued ==**
> >
> > **W3:** The experiments are not sufficient. The proposed method is very simple but experiments are only conducted on toy datasets(CIFA10) and outdated neural networks(AlexNet, ResNet18). More experiments on larger datasets(ImageNet) and modern neural networks(Vision Transformer) are encouraged.
> >
> > **A:** Thank you for your advice. This work aims to develop the proposed Assisted learning methods and validate their effectiveness in diverse ML tasks, e.g., deep learning and deep reinforcement learning, using various standard models, datasets and RL environments. Due to limited computation resource and time, conducting large scale experiments is not the focus of this paper.

---

### Comment · Reviewer_HsNu · 2022-12-06

Summary:

This paper considers a practically useful problem of training machine learning models with a small organization with limited & imbalanced data (they call the learner) and a large organization with sufficient & balanced data (they call the provider), in a collaborative but no-data-sharing manner. To solve this problem, propose two decentralized learning algorithms for deep learning tasks and reinforcement learning tasks respectively. They also provide some favorable results of the proposed methods.

Pros:

-The paper considers an interesting problem for two machine learning tasks, deep learning tasks, and reinforcement tasks. The experimental results are thorough.

Cons:

-The first concern is about the privacy leakage of the proposed method. It is known that model-inversion attacks can exploit machine-learning models to infer information about the training data and even reconstruct them. These attacks can proceed directly by analyzing the model parameters, but also indirectly by repeatedly querying the models. See papers:

The Secret Revealer: Generative Model-Inversion Attacks Against Deep Neural Networks, 2020

Reconstructing Training Data from Diverse ML Models by Ensemble Inversion, 2022

For the proposed method, in each communication round, a set of models need to be released to the other party, which imposes a great challenge on data privacy. Even though the proposed method does not share data directly, the sharing of models during training does share information about the data, in the worst case, they can even reconstruct each other’s data. This violates the original motivation of assisted learning.

-The second concern is that the assisted learning setting is very close to federated learning. From the reviewer’s perspective, the former can be a special case of the latter with two clients and non-IID data. In the paper, the authors discussed the difference as their framework requires a few interaction rounds between the learner and provider, which is a difference in the methods but not the problem setting itself.

-The third concern is about the assumptions of the assisted learning setting. As shown in the examples, the learner only has limited budget, e.g., a small local hospital, how can the learner afford communication and computation resources to the same level as the provider, e.g., a large medical center? Specifically, the learner model and provider model share the same architectures; during training, the learner and the provider need to exchange a set of models; to boost the performance, we may consider using very large deep neural networks, which may make the communication very costly, how to deal with these problems?

---

> ### Author Response · Authors · 2022-12-21
> **Thank you for the time and comments. We address each comment below.**
>
> **Q1:** It is known that model-inversion attacks can exploit machine-learning models to infer information about the training data. These attacks can proceed directly by analyzing the model parameters, but also indirectly by repeatedly querying the models.
> For the proposed method, in each communication round, a set of models need to be released to the other party, which imposes a great challenge on data privacy. Even though the proposed method does not share data directly, the sharing of models during training does share information about the data.
>
> **A:** We agree with the reviewer that data leakage is a fundamental problem in assisted learning as well as many other distributed learning frameworks (e.g., federated learning). We also thank the reviewer for pointing out the references on the model-inversion attack.
>
> While the focus of this work is not on privacy issues of assisted learning, we think many of the existing model-inversion defense strategies can be applied to address this issue in assisted learning. For example, the Mutual Information Regularization-based Defense (MID) proposed in [1] is a model-agnostic approach against model-inversion attacks. It restricts the attacker's ability to infer private training attributes by limiting the information about the model input contained in the prediction, and can be potentially applied to our assisted learning framework in the future. Another recent work [2] proposed an algorithm that trains an inversion model to mimic the attacker. Then a label modifier is applied to keep the label unchanged to avoid accuracy loss of the target model. This model-agnostic algorithm can also be applied to our framework because it does not tamper with the training process or need the private training dataset.
>
> [1] "Improving robustness to model inversion attacks via mutual information regularization". AAAI 2021.
>
> [2] "Defending against model inversion attack by adversarial examples". IEEE CSR 2021.
>
> **Q2:** Assisted learning setting is very close to federated learning. From the reviewer’s perspective, the former can be a special case of the latter with two clients and non-IID data. In the paper, the authors discussed the difference as their framework requires a few interaction rounds between the learner and provider, which is a difference in the methods but not the problem setting itself.
>
> **A:** Great comment. We agree with the reviewer that the standard federated learning algorithm can be applied to solve the same problem (which involves two clients), although it is originally designed for large-scale distributed systems. However, we note that the standard FedAvg algorithm is sensitive to data heterogeneity/imbalance. To illustrate this, we compare the performance of FedAvg with that of our AssistDeep in experiments with highly imbalanced data (see Appendix A.1 of the revision). It can be seen that FedAvg converges very slowly due to the highly imbalanced data, whereas our AssistDeep has much more robust performance. This is because FedAvg only linearly aggregates the models produced in the last iterations of local training, whereas AssistDeep performs screening over all the models produced in the local training and picks the best one. If the reviewer prefers, we can add FedAvg as a baseline for all the experiments of our paper.
>
> On the other hand, our AssistDeep algorithm is fundamentally different from FedAvg. To elaborate, in each round of FedAvg, the cloud server aggregates all the latest local models produced by the clients to obtain a global model. Such a strategy is sensitive to both local data heterogeneity and over-fitting in the local training. As a comparison, in our AssistDeep, both the learner and provider will send their local models and the corresponding loss values to the other side for screening, and the best local model that achieves the minimum global loss will be chosen as the initialization for the next round. Such a filtering process is highly nonlinear, as opposed to the linear weighted combination used in FedAvg.
>
> **Q3:** As shown in the examples, the learner only has a limited budget, e.g., a small local hospital, how can the learner afford communication and computation resources to the same level as the provider, e.g., a large medical center? Specifically, the learner model and provider model share the same architectures; during training, the learner and the provider need to exchange a set of models; to boost the performance, we may consider using very large deep neural networks, which may make the communication very costly, how to deal with these problems?
>
> **A:** Great question. This issue also arises in conventional federated learning, where the cloud can have much more resources than the clients. Below we provide some feasible solutions to address this issue in assisted learning.
>
> **=== To Be Continued (more responses are appended in the following text field) ===**

---

> > ### Author Response · Authors · 2022-12-21
> > **== Response Continued ==**
> >
> > First, note that in assisted learning, the data size of the learner is typically much smaller than that of the provider, and therefore a few local training iterations are usually sufficient for the learner. In fact, in all of our experiments, we set the total number of local training iterations in each round to be proportional to the local data size, and therefore the learner only needs a limited computation budget to support its local training.
> >
> > Second, in our setting, both the learner and provider are assumed to be organizations, not small devices as in federated learning. For example, it is reasonable to assume that a local hospital can budget a computing node with a standard GPU. The large amount of communication between the learner and the provider can be realized by, e.g., mailing hardrives to each other (fast and reliable), not necessarily through wireless communication over the cloud (can be very slow). This is acceptable as we have shown that assisted learning typically requires less than 10 communication rounds.
> >
> > On the other hand, to further reduce communication, it is possible to leverage some techniques such as knowledge distillation to enable using different models on both sides, i.e., the learner uses a small model whereas the provider uses a big model. Such a method has been proven to be very effective in the existing federated learning literature, see, e.g., the paper entitled ``Group Knowledge Transfer: Federated Learning of Large CNNs at the Edge''. Here in our assisted learning setting, both the learner and the provider may not want to share their own learning model architectures. To address this issue, they can agree to distill their local models to a common model, which will enable using small model on the learner side and using big model on the provider side. In our Appendix A.2, we provided one preliminary experiment that validates this idea, and we think this is an interesting direction that deserves comprehensive exploration in future study.

---

### Author Response · Authors · 2023-02-21
**== Response Continued from Our First Response to Reviewer gXtW ==**

1. (1) To answer this question, we modified the original AssistDeep algorithm to let the provider train on its local data using weights initialized from the model generated in the learner's last local training iteration (instead of the checkpoint model that achieves the minimum global loss). We name this algorithm as Revised-AssistDeep. Then under imbalanced and limited learner's data ($\gamma_L=0.3$) and different numbers of learner's local iterations ($T$ = 2000, 6000, 12000), we conducted experiments to train the AlexNet on CIFAR-10 dataset for both AssistDeep and Revised-AssistDeep to compare their performance. The experimental details and results are presented in Section A.5 in the Appendix of the revision. From all the results with different $T$ values (2000, 6000, 12000), it can be seen that Revised-AssistDeep performs worse than our AssistDeep. These results demonstrate that in the algorithm design of our AssistDeep, it is necessary for the provider to initialize its weights from the checkpoints passed by the learner to do the local training. \
(2) To answer this question, we conduct experiments on Provider-SGD by training AlexNet on both CIFAR-10 and SVHN datasets, and compare its performance with that of SGD, Learner-SGD and our AssistDeep under both imbalanced ($\gamma_{L}=0.5$) and balanced ($\gamma_{L}=1$) learner’s data. The experimental details and results are presented in Section A.6 in the Appendix of the revised paper. Both CIFAR-10 and SVHN results show that Provider-SGD performs worse than AssistDeep and SGD, but better than Learner-SGD. This proves that under both imbalanced and balanced learner’s data, our AssistDeep outperforms Provider-SGD since the former can leverage both the learner's and provider's data to improve the model's generalization performance while the latter leverages only the provider's data.

2. To address the issue of learner's data imbalance, we resample the data so that each batch will sample data points from each class with equal probability. Then we implemented the Learner-SGD with such a resampling strategy and named it Resampled-Learner-SGD. We compare it with the original Learner-SGD and our AssistDeep in training different models on different datasets. The experiments and results are added to Section A.7 in the Appendix of the revised paper. From all the results, it can be seen that the performance of Resampled-Learner-SGD is very similar to that of Learner-SGD and worse than AssistDeep, which proves the advantage of our AssistDeep over the Resampled-Learner-SGD.

3. Actually in our paper, the parameter $\gamma_{L}$ already controls the relative data size of the learner over the provider, so the results of different relative data sizes have already been presented in Section 4.2.1. To explain more specifically, in our paper the hyper-parameter $\gamma_{L}$ controls both learner's data size and data heterogeneity. Referring to the definition of $\gamma_{L}$ described in Section 4.2,  we know that the smaller the $\gamma_{L}$, the smaller the learner's data size, and the more imbalanced the learner's data. For example, $\gamma_{L}=1$ means that the learner's data is balanced and large whereas $\gamma_{L}=0$ means that the learner's data is extremely imbalanced and small. Given that we have fixed the provider's data size and used $\gamma_{L}$ to vary the learner's data size in our paper, we have already presented the results with the different relative data sizes of the learner over the provider.

**Q4:**

**Weaknesses:** AssistDeep requires specifying T and T' for the training duration of the client and provider models as well as standard hyperparameters for the model. It is unclear how to tune hyperparameters in this setting.

**Requested Changes:** Please discuss how sensitive AssistDeep is to T and T' and how you would go about tuning hyperparameters for the model on both client and provider.

**A:** We have conducted additional experiments with different models and training data to test the sensitivity of AssistDeep to $T$ and $T'$. Note that in our paper, we fix the total number of iterations $T+T'=2000$ by default and set $T$ and $T'$ in proportion to the local data sizes of the learner and provider. Thus, in the additional experiments, we consider several different values of $T+T'=500, 1000, 2000, 4000$, and the comparison results are shown in Section A.8 in the Appendix of the revised paper. From the results, one can see that AssistDeep achieves very similar performance under different values of $T$ and $T'$, indicating that our AssistDeep is not sensitive to this hyper-parameter. Consequently, the other hyper-parameters tuned in our default setting $T+T'=2000$ can also be applied to other settings of $T+T'$.

---

### Decision · Action_Editors · 2023-04-23

**Recommendation:** Accept with minor revision

**Comment:**

The authors have done a good job in revising their paper and addressing the review comments. Most of the concerns raised by the reviewers have been addressed in some way. However, it seems that people still have problems with the relationship (or distinction) between assisted learning and federated learning. I hope the authors can make a systematic discussion on this topic in their future revision. With a better positioning, this work will demonstrate clearer value to the community.

**Audience:**

Yes

**Claims And Evidence:**

Yes